# `HieraMAS`: Optimizing Intra-Node LLM Mixtures and Inter-Node Topology for Multi-Agent Systems

Tianjun Yao [* 1 2]   Zhaoyi Li [* 2]   Zhiqiang Shen [2]

## Abstract

Multi-agent systems (MAS) built on large language models (LLMs) have demonstrated remarkable performance across diverse tasks. Existing approaches optimize communication topology, role assignment, or LLM routing in isolation, while treating each agent as a monolithic unit—failing to exploit internal LLM mixtures that can enhance individual role capabilities. We propose `HieraMAS`, a hierarchical agent collaboration framework with intra-node LLM mixtures and inter-node communication topology. HieraMAS introduces *supernodes*, where each functional role comprises multiple heterogeneous LLMs in a propose-synthesis structure. The optimization of `HieraMAS` poses unique credit assignment challenges, as final task performance heavily depends on LLM capabilities, potentially causing erroneous reinforcement of suboptimal configurations. We address this via a two-stage algorithm: (1) multi-level reward attribution providing fine-grained feedback at both node and system levels; and (2) graph classification treating topology selection as a holistic task rather than per-edge optimization. Experiments on reasoning and coding benchmarks demonstrate that `HieraMAS` significantly outperforms existing methods while achieving better cost-performance trade-offs.[1]

## 1. Introduction

Recent advances in large language model (LLM) based agents have revealed a parallel phenomenon in artificial intelligence. A growing body of research demonstrates that multi-agent systems (MAS) substantially outperform single-agent approaches across diverse tasks (Du et al., 2024a; Liang et al., 2024; Talebirad & Nadiri, 2023; Chen et al., 2024), catalyzing the development of numerous MAS frameworks (Wu et al., 2024; Hong et al., 2024; Qian et al., 2024). Beyond application-specific designs, researchers have identified fundamental challenges in multi-agent coordination and proposed systematic optimization approaches along several key dimensions: communication topology learning (Zhuge et al., 2024; Zhang et al., 2024a; 2025), which optimizes the information flow structure between agents; role assignment and specialization (Qian et al., 2025b; Hong et al., 2024; Liu et al., 2023b), which determines how agents are assigned distinct functional responsibilities; and LLM routing (Ong et al., 2024; Yue et al., 2025), which selects appropriate backbone models for different agent roles to balance cost and capability. Concurrently, researchers have discovered that LLMs generate substantially better responses when provided with outputs from other models as auxiliary input (Wang et al., 2024; Li et al., 2025a;b). This phenomenon offers an intriguing connection to MAS: MAS inherently involves collaboration among potentially heterogeneous LLMs through different roles and communication patterns, while the aforementioned phenomenon provides an alternative form of LLM collaboration through input-output composition. However, this connection remains unexplored in current research. Existing approaches either focus on role assignment (Liu et al., 2023b; Chen et al., 2023) or LLM routing (Yue et al., 2025), but none effectively integrates the collaborativeness property, as they treat each agent as a monolithic unit rather than exploiting the potential for internal LLM mixtures to enhance individual role capabilities.

In this work, we propose `HieraMAS` (**Hiera**rchical Collaboration with **MAS**), which unifies these two forms of collaboration within a coherent MAS framework. Unlike conventional MAS where each node corresponds to a single agent, `HieraMAS` introduces *supernodes*, each comprising multiple potentially heterogeneous agents that implement a propose-synthesis structure to augment each functional role. Intuitively, enhancing each role's capability through internal mixtures may induce cascading effects on the overall system, e.g., some communication edges may no longer be necessary, leading to sparser and more efficient structures. Motivated by this hypothesis, `HieraMAS` jointly optimizes

---

[*]Equal contribution [1]Shenzhen Loop Area Institute, Shenzhen, China. [2]Mohamed bin Zayed University of Artificial Intelligence, Abu Dhabi, UAE.

*Proceedings of the $43^{rd}$ International Conference on Machine Learning*, Seoul, South Korea. PMLR 306, 2026. Copyright 2026 by the author(s).

[1]Code can be found at: https://github.com/tianyao-aka/HieraMAS

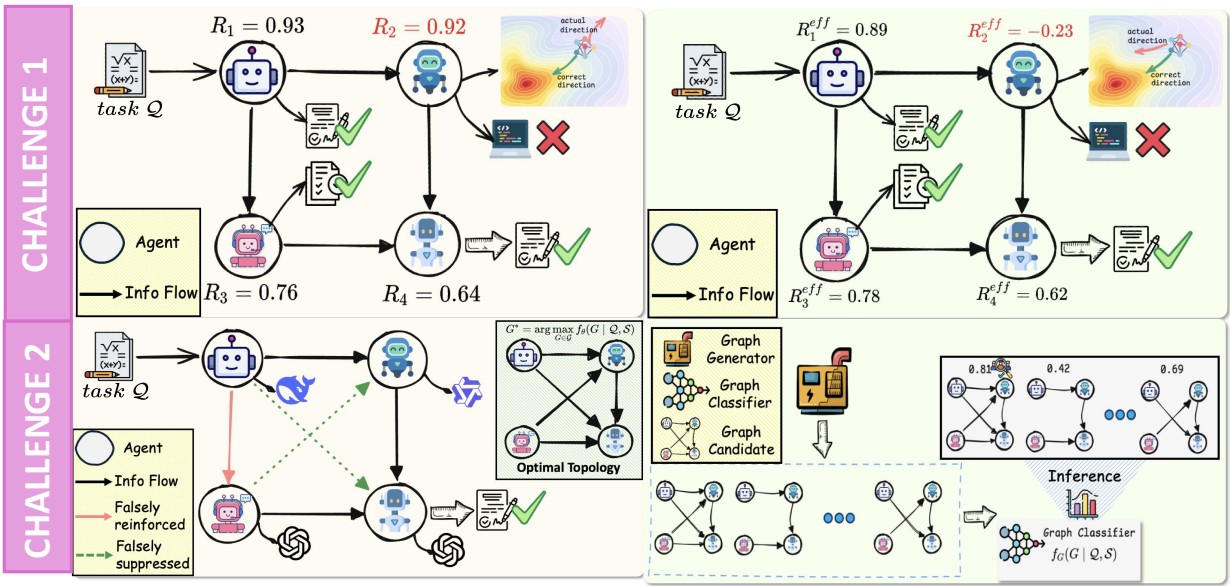

*Figure 1.* Illustration of two credit assignment challenges in joint optimization and our solutions. **Challenge 1**: Final task rewards mask individual node errors—Node 2 produces incorrect output but receives high reward $R_2 = 0.92$. **HieraMAS** addresses this via multi-level rewards that provide effective per-role attribution ($R_2^{eff} = -0.23$). **Challenge 2**: Per-edge optimization suffers from entangled attribution, where edges may be falsely reinforced or suppressed. **HieraMAS** reformulates topology selection as a holistic graph classification task, using a graph generator to produce candidates and a graph classifier to select the optimal topology.

three interconnected dimensions: ① graph topology that determines inter-supernode communication patterns; ② role pruning that identifies which functional roles to retain; and ③ LLM selection within supernodes that configures the internal agent mixtures. This joint optimization can naturally be formulated as a credit assignment problem (Sutton, 1984) using reinforcement learning (Sutton et al., 1998). However, compared to methods that optimize a single dimension, credit assignment becomes substantially more difficult in our setting (Figure 1). ① Relying solely on final task rewards leads to inaccurate per-role attribution, as high rewards may mask individual node errors that were compensated by other agents. ② Per-edge optimization faces similar attribution challenges but is even more challenging, as the contribution of individual communication edges is entangled with both the sending and receiving nodes' behaviors, making it challenging to isolate edge-level effects. To address these challenges, **HieraMAS** employs a two-phase algorithm: in the first phase, we use multi-level rewards rather than final outcome rewards alone, providing fine-grained attribution signals for optimizing supernode (Def. 1) configurations; in the second phase, given the inherent difficulty of edge-level credit assignment, we propose treating topology selection as a holistic graph classification task rather than per-edge optimization. We summarize our contributions as follows:

• We introduce a novel MAS paradigm where collaborativeness emerges at two levels: *intra-node* collaboration through internal LLM mixtures within supernodes, and

*inter-node* collaboration through communication across functional roles.

• We propose **HieraMAS**, a unified framework that jointly optimizes intra-node configurations (LLM selection and role retention) and inter-node structures (communication topology), enabling holistic system optimization.

• We design multi-level rewards for fine-grained per-role credit assignment, and reformulate topology optimization as a graph classification task to circumvent the intractable edge-level attribution problem.

• We conduct extensive experiments on programming, mathematical reasoning, and general knowledge benchmarks spanning diverse subjects. **HieraMAS** achieves state-of-the-art performance while maintaining cost-efficiency.

## 2. Preliminaries

In this section, we formalize the **HieraMAS** framework as a Markov Decision Process (MDP) and introduce its optimization objectives.

### 2.1. Notation Establishment

**Search Space.** We define the search space of a MAS as $\mathbb{S} = (\mathbb{M}, \mathbb{R}, \mathbb{G})$, where $\mathbb{M}$ denotes the pool of $N_m$ available LLM backbones (including a special `skip` token), $\mathbb{R}$ represents the set of $N_r$ predefined agent roles (e.g., Mathematical Analyst, Math Solver, Inspector), and $\mathbb{G}$ denotes the space

of graph topologies encoding inter-agent communication (e.g., Chain, FullConnected, Star).

**Definition 1** (Supernode). A *supernode* $S_i$ is a mixture-of-LLMs unit with role $r_i \in \mathbb{R}$, consisting of $W$ proposer positions and one synthesizer:

$$S_i = \left( r_i, \{m_{i,j}^{(w)}\}_{j=1}^W, m_i^{(a)} \right), \quad m_{i,j}^{(w)}, m_i^{(a)} \in \mathbb{M}. \quad (1)$$

Here, $i$ indexes the supernode within the system, $j \in \{1, \ldots, W\}$ indexes proposer positions within supernode $S_i$, the superscript $(w)$ denotes proposer LLMs that generate diverse proposals, and $(a)$ denotes the synthesizer LLM that synthesizes proposer outputs into a unified response. Internally, each proposer connects to the synthesizer, which aggregates the diverse proposals from proposers and generates the final response. This internal structure is inspired by prior work (Wang et al., 2024) and is not optimized.

## 2.2. MDP Formulation

We formulate **HieraMAS** as an MDP $(\mathcal{X}, \mathcal{A}, P, R)$:

**State $\mathcal{X}$.** The state encodes the current configuration of the MAS:

$$x = \left( \{r_i\}_{i=1}^N, \{m_{i,j}^{(w)}, m_i^{(a)}\}_{i,j}, \mathbf{E}, \mathcal{Q} \right), \quad (2)$$

where $\{r_i\}_{i=1}^N$ are the role assignments for up to $N$ supernodes, $\{m_{i,j}^{(w)}, m_i^{(a)}\}$ are the LLM assignments within each supernode, $\mathbf{E} \in \{0,1\}^{N \times N}$ is the adjacency matrix, and $\mathcal{Q}$ is the input query.

**Action $\mathcal{A}$.** The action space consists of three components: (1) *Role selection*: for each supernode $S_i$, select a role $r_i \in \mathbb{R}$ or deactivate it; (2) *LLM selection*: for each position within a supernode (up to $W + 1$ positions), select an LLM $m \in \mathbb{M}$ or $\texttt{skip}$; and (3) *Edge selection*: select a subset of edges $\mathbf{E} \subseteq \mathbb{G}$ to enable communication among supernodes.

**Transition $P$.** Given state $x$ and action $a$, the transition $P(x'|x, a)$ deterministically updates the system configuration and executes the MAS to obtain outputs.

**Reward $R$.** The reward balances task performance and computational cost:

$$R(x, a) = f\left( U(\mathcal{S}; \mathcal{Q}, a^*), C(\mathcal{S}; \mathcal{Q}) \right), \quad (3)$$

where $\mathcal{S} = \{S_i\}_{i=1}^N$ denotes the set of all supernodes in the system, $U(\cdot)$ measures correctness against ground-truth $a^*$, and $C(\cdot)$ quantifies token expenditure.

## 2.3. Optimization Objective

Given a dataset $\mathcal{D}$ of queries $\mathcal{Q}$ with ground-truth answers $a^*$, **HieraMAS** aims to learn a policy $\pi_\theta$ that maximizes expected reward:

$$\max_\theta \mathbb{E}_{(\mathcal{Q}, a^*) \sim \mathcal{D}, a \sim \pi_\theta(\cdot|x)} \left[ R(x, a) \right]. \quad (4)$$

## 3. Method

In this section, we introduce our **HieraMAS** framework. Given a query $\mathcal{Q}$, our framework progressively constructs a customized MAS by: (1) selecting optimal LLMs for each position within supernodes, and (2) selecting an appropriate inter-supernode communication topology. We employ a two-stage training algorithm that addresses the unique credit assignment challenges arising from joint optimization.

## 3.1. LLM Selection within Supernodes

Each supernode $S_i$ contains $W$ proposer positions and one synthesizer position. We learn to select the optimal LLM for each position based on task characteristics and role requirements. We encode the query with role information as $\mathbf{h}_{\mathcal{Q},r} = f_\psi(\mathcal{Q}, r, \text{desc}(r))$ using a sentence encoder (Reimers & Gurevych, 2019), and pre-compute LLM profile embeddings $\mathbf{h}_{m_\ell} = f_\psi(\text{profile}(m_\ell))$ for each $m_\ell \in \mathbb{M}$. The selection probability over LLMs is:

$$\pi_m(m_\ell|\mathcal{Q}, r) = \frac{\exp(s_\ell/\tau)}{\sum_{m' \in \mathbb{M}} \exp(s_{m'}/\tau)}, \quad (5)$$
$$\text{where} \quad s_\ell = \text{MLP}(\mathbf{h}_{\mathcal{Q},r}, \mathbf{h}_{m_\ell})$$

and $\tau$ is the temperature.

**Role and proposer Pruning via Skip Token.** Critically, $\mathbb{M}$ includes a special $\texttt{skip}$ token that enables automatic pruning at multiple granularities. When $\texttt{skip}$ is selected for a proposer position, that proposer is omitted and incurs zero cost, allowing the system to adaptively reduce the mixture size within a supernode. When $\texttt{skip}$ is selected for an synthesizer position, the *entire supernode* is deactivated, effectively pruning that functional role from the MAS. This unified mechanism enables **HieraMAS** to jointly learn both the optimal LLM configuration and which roles/proposers are necessary for a given query, without requiring separate pruning modules.

## 3.2. Graph Topology Selection

To circumvent the challenge of the per-edge credit assignment, we propose treating topology selection as a *holistic graph classification problem* rather than per-edge optimization. Instead of learning edge probabilities directly, we train a graph classifier that scores a pool of topology candidates and selects the most suitable one for each query. This formulation sidesteps edge-level credit assignment by evaluating topologies as indivisible units.

**Graph Candidate Pool.** We pre-generate a diverse pool of $K$ random directed acyclic graphs (DAGs) $\mathcal{G} = \{G_1, G_2, \ldots, G_K\}$, where each graph $G_k$ represents a potential communication topology. The graphs are sampled with varying edge densities to ensure diversity, covering sparse, medium, and dense connectivity patterns.

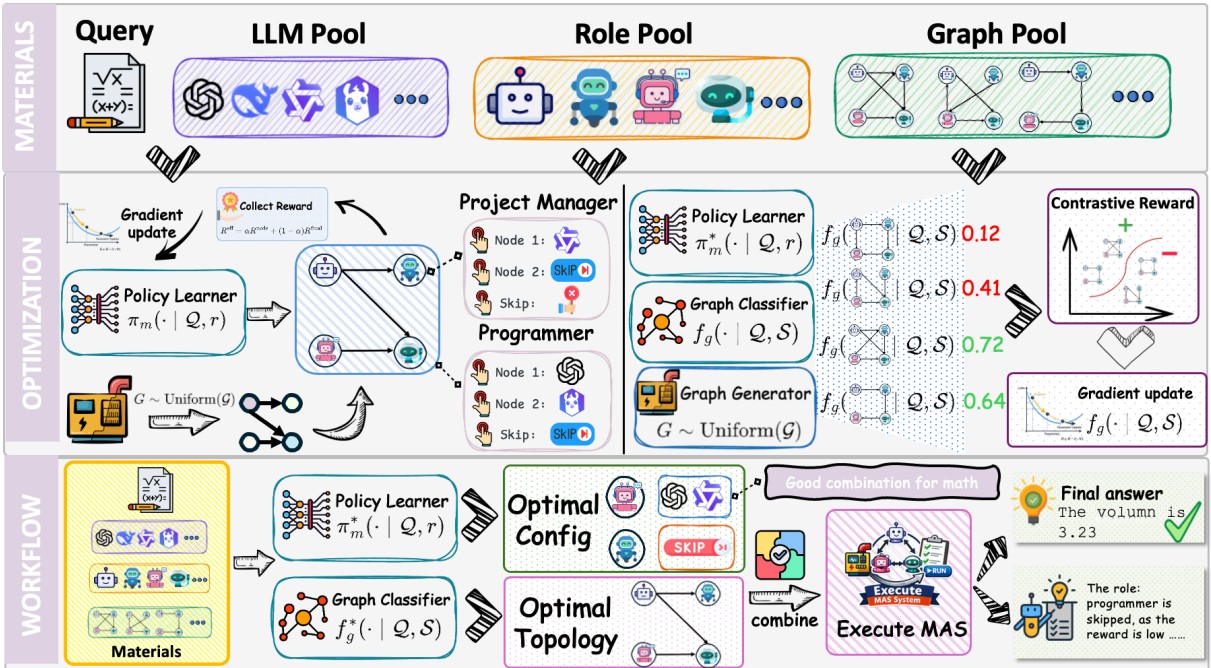

Figure 2. The overall framework of `HieraMAS`. By optimizing a policy learner $\pi_m$ with multi-level rewards (Stage 1) and a graph classifier $f_G(\cdot)$ with contrastive rewards (Stage 2), `HieraMAS` learns to select optimal supernode configurations and communication topologies. During inference, the trained modules jointly determine the supernode configurations and graph topology, then execute the MAS to produce the final answer.

**Graph Classifier.** Given a query $\mathcal{Q}$ and a candidate graph $G_k$ represented by its adjacency matrix $\mathbf{A}_k \in \{0,1\}^{N \times N}$, the classifier predicts a suitability score. We first construct node features by concatenating role embeddings with the query embedding, then apply a Graph Convolutional Network (GCN) (Kipf & Welling, 2017) to obtain graph-aware node representations:

$$\mathbf{Z} = \text{GCN}(\mathbf{X}, \mathbf{A}_k), \quad \mathbf{z}_G = \text{Pool}(\mathbf{Z}), \quad s_k = \text{MLP}(\mathbf{z}_G, \mathbf{h}_{\mathcal{Q}}), \tag{6}$$

where $\mathbf{X}$ contains node features and $\text{Pool}(\cdot)$ aggregates node representations into a graph-level embedding. At inference, we select $G^* = \arg\max_{G_k \in \mathcal{G}} s_k$.

### 3.3. Two-Stage Training Algorithm

Training `HieraMAS` requires addressing the entangled optimization of supernode configurations and graph topology. We propose a two-stage algorithm that decouples these components to enable more effective learning. The complete procedure is summarized in Algorithm 1.

#### 3.3.1. STAGE 1: SUPERNODE OPTIMIZATION WITH RANDOM GRAPHS

In the first stage, we focus on learning optimal LLM selection within supernodes while using randomly sampled graph topologies. For each training sample, we randomly select a graph $G_k$ from the candidate pool $\mathcal{G}$ and execute the MAS with this fixed topology. This design serves two purposes: (1) it exposes the LLM selector to diverse communication patterns, preventing overfitting to a single topology; and (2) it decouples supernode learning from topology learning, allowing the selector to learn robust LLM assignments that generalize across different graph structures.

**Multi-Level Reward.** To address the per-node credit assignment challenge (Challenge 1 in Figure 1), we employ a multi-level reward structure that provides feedback at both the system level and individual supernode level. For each supernode $S_i$, we compute a node-level reward $R_i^{\text{node}}$ by evaluating its synthesizer output against the ground truth. The effective reward combines both signals:

$$R_i^{\text{eff}} = \alpha \cdot R_i^{\text{node}} + (1 - \alpha) \cdot R^{\text{final}}, \tag{7}$$

where $\alpha \in [0, 1]$ controls the mixing coefficient and $R^{\text{final}}$ is the reward from the decision node's output. For roles whose outputs are not directly comparable to the final answer (e.g., planners, critics), we set $\alpha = 0$ to rely solely on the final reward signal.

**Cost-Aware Reward Function.** Each reward (both $R_i^{\text{node}}$ and $R^{\text{final}}$) is computed using a cost-sensitive reward func-

tion that balances correctness and computational cost:

$$R(u, C) = \begin{cases} \exp(-\lambda \cdot C), & \text{if } u = 1 \\ -\exp(\lambda \cdot C), & \text{if } u = -1 \end{cases} \quad (8)$$

where $u \in \{-1, 1\}$ is the utility (correctness), $C$ is the token cost, and $\lambda$ is the cost sensitivity hyperparameter. This formulation encourages the policy to select cost-efficient LLMs: correct solutions yield positive rewards that decrease with cost, while failures incur negative rewards that grow more severe with higher cost.

**Stage 1 Training Objective.** The total loss for Stage 1 combines policy gradient with entropy regularization to encourage exploration:

$$\mathcal{L}_{\text{stage1}} = -\sum_{i=1}^{N} \log \pi_m(\mathbf{m}_i | \mathcal{Q}, r_i) \cdot R_i^{\text{eff}} - \lambda_H \sum_p H(\pi_m^{(p)}),$$
$$(9)$$

where $H(\pi_m^{(p)}) = -\sum_{m_\ell} \pi_m(m_\ell) \log \pi_m(m_\ell)$ is the entropy of LLM selection at position $p$, preventing premature convergence to suboptimal configurations.

### 3.3.2. STAGE 2: GRAPH CLASSIFIER TRAINING

After Stage 1 converges, we freeze the LLM selector and train the graph classifier. This stage generates labeled training data by executing the fixed MAS with different graph topologies.

**Data Generation.** For each task $\mathcal{Q}$ in the training set, we sample $M$ random graphs from $\mathcal{G}$ and execute the MAS with each topology using the frozen LLM selector. We record the reward $R_k$ for each graph $G_k$, and label the top-performing graphs (with positive reward) as positive examples.

**Classifier Training.** We train the graph classifier using binary cross-entropy loss:

$$\mathcal{L}_{\text{stage2}} = \text{BCE}(y_k, \sigma(s_k)), \quad (10)$$

where $y_k$ is the label, $\sigma(\cdot)$ is the sigmoid function.

This two-stage approach offers several advantages: (1) it avoids the credit assignment problem in edge-level optimization by treating topologies holistically; (2) it leverages the optimized LLM selector to generate meaningful training signals for the graph classifier; and (3) it enables efficient inference by simply scoring pre-generated graph candidates rather than sampling edges stochastically.

### 3.4. Theoretical Analysis

We provide theoretical justification for our two-stage design in addressing the credit assignment challenges.

**Theorem 3.1.** *Consider optimizing a multi-agent system with $N$ supernodes and a communication graph $G \in \mathcal{G}$.*

*(i) (**Per-node credit assignment**) Under final-reward-only training, when a failing supernode's error is compensated by other agents, its policy gradient points in the wrong direction. Multi-level rewards with sufficient weight on node-level feedback ensure the gradient sign matches the desired update direction.*

*(ii) (**Per-edge credit assignment**) Per-edge policy gradient optimization incurs an irreducible error rate of $\Omega((1-\rho)q)$, where $\rho$ is the fraction of optimal edges and $q$ is the probability of high final reward. In contrast, holistic graph scoring with random graph generation reduces this to a vanishing estimation error $\mathcal{O}(1/\sqrt{N_{samples}})$, enabling correct topology identification with sufficient samples.*

Intuitively, multi-level rewards prevent the masking effect where system-level success hides individual failures, while graph-level scoring transforms the ill-posed per-edge credit assignment into a well-posed estimation problem. The formal proofs are deferred to Appendix C.

## 4. Experiments

### 4.1. Experimental Setup

**Datasets and Metrics.** We evaluate our approach on three diverse benchmark datasets to comprehensively assess its performance across different task types: **(1) HumanEval++** (Liu et al., 2023a): An enhanced version of the original HumanEval benchmark (Chen, 2021), featuring more robust evaluation via improved test suites for function-implementation tasks in code generation. For all benchmarks, we report Pass@1 (%) as the primary evaluation metric. **(2) MATH** (Hendrycks et al., 2021): A mathematical reasoning benchmark containing challenging high school competition problems requiring multi-step reasoning. **(3) MMLU-Redux** (Gema et al., 2025): A decontaminated and disambiguated subset of the original MMLU benchmark (Hendrycks et al., 2020), covering 30 subjects across STEM, humanities, and social sciences. For all benchmarks, we report accuracy (%) as the primary evaluation metric.

**Baselines.** We compare against a comprehensive set of baselines spanning single-agent and multi-agent approaches: **Single-agent methods**: Base (direct prompting), Chain-of-Thought (CoT) (Wei et al., 2022). **Fixed multi-agent methods**: Self-Consistency (Wang et al., 2022), Self-Consistency+CoT, LLM-Debate (Du et al., 2024b; Liang et al., 2024), Full-Graph (fully-connected topology with all agents communicating), and Random-Graph (randomly generated communication topology). **Learning-based multi-agent methods**: AFlow (Zhang et al., 2024b), which optimizes workflows using Monte Carlo Tree Search; GDesigner (Zhang et al., 2024a), which uses GNN-based topol-

*Table 1.* Main results on three benchmarks. We report accuracy (%) for each dataset. ✓ indicates the method uses the corresponding component: **Multi** = multi-agent, **Topo** = optimizing topology, **Role** = optimizing roles and LLMs in each role, **Node** = optimizing intra-node configuration. Best results are in **bold**, second best are underlined.

| Method | Multi | Topo | Role | Node | HumanEval++ | | MATH | | MMLU-Redux | | Avg. |
|---|---|---|---|---|---|---|---|---|---|---|---|
| | | | | | GPT-5-Mini | Qwen3-80B | GPT-5-Mini | Qwen3-80B | GPT-5-Mini | Qwen3-80B | |
| Base | ✗ | ✗ | ✗ | ✗ | 89.06 | 84.14 | 77.78 | 74.44 | 92.00 | 82.40 | 83.14 |
| CoT | ✗ | ✗ | ✗ | ✗ | 87.50 | 85.94 | 92.22 | 90.00 | 93.60 | 89.60 | 89.81 |
| Self-Consistency | ✓ | ✗ | ✗ | ✗ | 89.06 | 87.50 | 93.33 | 91.11 | 94.40 | 83.20 | 89.77 |
| Self-Consistency+CoT | ✓ | ✗ | ✗ | ✗ | 90.62 | 85.94 | 94.44 | 92.22 | 93.60 | **92.80** | 91.60 |
| LLM-Debate | ✓ | ✗ | ✗ | ✗ | 87.50 | 87.50 | 94.44 | 94.44 | 92.80 | 92.00 | 91.45 |
| Full-Graph | ✓ | ✗ | ✗ | ✗ | 89.06 | 92.19 | 95.56 | **96.67** | 94.40 | 88.80 | 92.78 |
| Random-Graph | ✓ | ✗ | ✗ | ✗ | 85.94 | 92.19 | 93.33 | 94.44 | 91.20 | 88.00 | 90.85 |
| AFlow | ✓ | ✓ | ✗ | ✗ | 95.31 | **98.44** | 95.56 | 84.44 | 91.20 | 91.20 | 92.69 |
| GDesigner | ✓ | ✓ | ✗ | ✗ | 90.62 | 93.75 | 91.11 | 87.77 | 92.00 | 88.80 | 90.68 |
| MASRouter | ✓ | ✓ | ✓ | ✗ | **96.88** | **98.44** | 91.11 | 88.88 | 88.33 | 81.67 | 90.89 |
| **Ours** | ✓ | ✓ | ✓ | ✓ | 93.75 | 96.88 | **96.67** | 95.56 | **95.20** | 89.60 | **94.61** |

ogy optimization; and MASRouter (Yue et al., 2025), which learns to route between predefined topologies.

**LLM Pools.** We use a diverse pool of LLMs for heterogeneous agent assignment: Qwen3-8B (Yang et al., 2025), Qwen3-Next-80B-A3B-Instruct (Yang et al., 2025), DeepSeek-R1-Distill-Qwen-14B (Guo et al., 2025), Llama-3.1-8B-Instruct (Dubey et al., 2024), DeepSeek-V3.2 (DeepSeek-AI, 2024), and Gemma-3-27B-IT (Team et al., 2025), GPT-5-Mini (OpenAI, 2025), GPT-5-Nano (OpenAI, 2025) and GPT-4o-Mini (Achiam et al., 2023). All models are accessible via open APIs, supporting reproducibility of our experiments. We conduct experiments with 2 settings: GPT-5-Mini and Qwen3-Next-80B-A3B-Instruct. For the GPT-5-Mini setting, methods that support learnable LLM selection (**HieraMAS** and MASRouter) use all LLMs in the pool; for the Qwen3-80B setting, these methods exclude all GPT models from the pool. For other methods, GPT-5-Mini and Qwen3-Next-80B-A3B-Instruct are utilized respectively.

**Implementation Details.** Details on implementations of **HieraMAS** are deferred in Appendix D.1.

## 4.2. Main Results

Table 1 presents the main results comparing our approach against all baselines. Our method achieves the best average performance of **94.61%**, outperforming AFlow and MAS with full graph by a large margin. Notably, our approach demonstrates consistent improvements across all three benchmarks, achieving the best performance on HumanEval++ and MATH, and competitive results on MMLU-Redux. ① Full-Graph achieves 92.78% average accuracy by enabling all agents to communicate, but incurs significant computational overhead, for instance, on MMLU-Redux with GPT-5-Mini, our method costs $1.29 while Full-Graph requires $4.23 (**3.27× more expensive**). AFlow op-

*Table 2.* Ablation study on MATH and MMLU-Redux benchmarks. We evaluate the contribution of graph topology scoring and LLM selection. Acc (%) and Cost (USD) are reported.

| Variant | MATH | | MMLU-Redux | |
|---|---|---|---|---|
| | Acc | Cost | Acc | Cost |
| w/o Graph | 93.33 | 1.57 | 92.00 | 1.36 |
| w/o LLM Selection | **97.78** | 2.56 | 94.40 | 3.92 |
| **Ours** | 96.67 | **1.52** | **95.20** | **1.29** |

timizes workflows via MCTS with Claude 3.5-Sonnet (Anthropic, 2024), resulting in prohibitive training costs: on HumanEval++ with GPT-5-Mini, our training cost is 18.41x cheaper than AFlow (More training cost is included in Appendix D). ② LLM-Debate and Random-Graph rely on fixed debate patterns or random structures, trailing our approach by over 3%. ③ GDesigner and MASRouter both employ learning-based topology optimization. However, MASRouter routes between only 4 predefined regular topologies (e.g., debate, chain), constraining its search space. Our method instead learns to score and select from diverse random graph structures, discovering more effective collaboration strategies that better exploit heterogeneous LLM capabilities. This gap also highlights the importance of learned topologies, consistent with recent findings that irregular topologies outperform regular ones (Qian et al., 2025a).

## 4.3. In-depth Analysis

We conduct more experimental analysis in this section, using GPT-5-Mini setting. Additional experimental results are demonstrated in Appendix D, including: (1) The effect of #Proposers in supernodes, and (2) The effect of the size of graph pool $\mathcal{G}$.

**Ablation Study.** We conduct ablation experiments to evaluate the contribution of each component in our framework,

as shown in Table 2. We consider two variants: (1) *w/o Graph*, which removes the graph scoring mechanism and instead uses three randomly sampled graphs with averaged outputs; (2) *w/o LLM Selection*, which uses the learned optimal graph structure $\mathcal{G}^*$ but assigns **all agents the strongest LLM backbone** (i.e., GPT-5-Mini).

Removing the graph scoring mechanism (*w/o Graph*) leads to substantial performance degradation on both benchmarks: **3.45%** drop on MATH and **3.36%** drop on MMLU-Redux, demonstrating the importance of learned topology selection. For *w/o LLM Selection*, using only the learned structure $\mathcal{G}^*$ with **a fixed strongest LLM (GPT-5-Mini)** yields marginal improvement on MATH, but at the expense of significantly higher cost ($2.56 vs. $1.52). On MMLU-Redux, using GPT-5-Mini for all nodes incurs 203.9% more cost while achieving lower accuracy compared to **HieraMAS**. This demonstrates that our heterogeneous LLM assignment strategy in the supernode, mixing weaker and stronger models based on role requirements, achieves better cost-performance trade-offs than uniformly using the strongest LLM in the MAS.

**Analysis on Learned Topology in MMLU-Redux.** We visualize the most frequently selected communication topologies learned by **HieraMAS** on MMLU-Redux under GPT-5-Mini setting in Figure 3. The pairwise Jaccard similarity between the top-5 graphs ranges from 0.11 to 0.44, indicating substantial structural diversity in the learned topologies. Despite this diversity, we identify several consistent patterns across the top-ranked graphs.

*Common structural properties:* (1) **Sink nodes**: The Psychologist and Doctor roles consistently serve as sink nodes (high in-degree, zero out-degree) across most top graphs, receiving information from multiple sources but not propagating further. This suggests these roles function as final synthesizers or decision-makers. (2) **Source nodes**: The Critic role consistently acts as a primary source node (high out-degree, low in-degree), broadcasting information to multiple agents. The Economist also frequently serves as a hub with high out-degree. (3) **Sparse and irregular structure**: The density ratios range from 0.23 to 0.32, indicating that the learned topologies are considerably sparser than a fully-connected graph (density=1.0).

**Analysis on Intra-Node Configuration.** We analyze the dataset-level LLM selection preferences $\pi_m$ learned by **HieraMAS** for proposer and synthesizer positions within supernodes, as shown in Figure 4. The normalized logits reveal that **HieraMAS** learns distinct LLM preferences tailored to different task characteristics. For proposer nodes, although the preferred LLMs vary across tasks, they consistently form **strong-weak combinations** that balance capability and cost. On HumanEval++, Llama-3.1-8B-Instruct, DeepSeek-R1-Distill-Qwen-14B, and DeepSeek-

*Table 3.* Generalization performance (%) on unseen MMLU categories. Best in **bold**, second best underlined.

| Method | GPT-5-Mini | Qwen3-80B |
|---|---|---|
| SC+CoT | 68.00 | 64.00 |
| AFlow | **72.00** | 68.00 |
| GDesigner | **72.00** | 68.00 |
| MASRouter | 56.00 | 68.00 |
| **Ours** | 68.00 | **72.00** |

V3.2 emerge as the most preferred models. For MATH, Qwen3-Next-80B-A3B-Instruct and Qwen3-8B dominate the selection, while MMLU-Redux favors GPT-5-Nano and GPT-5-Mini. These patterns reveal task-specific model strengths: the GPT-5 series excels on general knowledge tasks requiring broad coverage, whereas Qwen3-Next-80B-A3B-Instruct offers superior cost-effectiveness for mathematical reasoning. For synthesizer nodes, we observe highly consistent preferences with proposer nodes, suggesting that LLM selection is primarily driven by task characteristics rather than positional roles within the supernode architecture. Notably, the skip token is rarely selected across all tasks, indicating that **HieraMAS** prefers to reduce costs through sparser communication topologies (density 0.23-0.32) rather than removing functional roles from the MAS.

**Generalization Analysis on MMLU-Redux.** We evaluate out-of-domain generalization by reserving 5 subjects from MMLU-Redux that are excluded from both training and in-domain testing, using the remaining 25 subjects for training, as shown in Table 3. GDesigner and AFlow, despite being learning-based methods, demonstrate competitive generalization, comparable to or exceeding SC+CoT. This may be attributed to their use of the strongest available LLM during execution without learning, which provides an implicit bias toward generalization. In contrast, MAS-Router and **HieraMAS** involve more extensive learning components (LLM/Role selection and topology optimization), posing greater challenges for out-of-domain generalization. Notably, MASRouter exhibits significant degradation under GPT-5-Mini setting, as its most frequently selected LLMs are Qwen3-Next-80B-A3B-Instruct, Qwen3-8B, and Gemma-3-27B, models that may not generalize well to unseen subjects. In comparison, **HieraMAS** maintains performance by preferring GPT-5-Mini and GPT-5-Nano, which are more amenable to generalization across diverse subjects due to their broad pretraining coverage. Additionally, we hypothesize that the learned communication topologies in **HieraMAS**, with Psychologist and Doctor as sink nodes and Critic as the source node, represent more generalizable collaboration patterns that transfer effectively to unseen domains.

**Cost Analysis.** We compare inference costs with MAS-

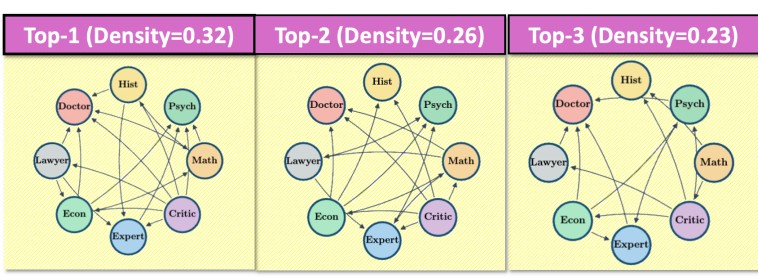

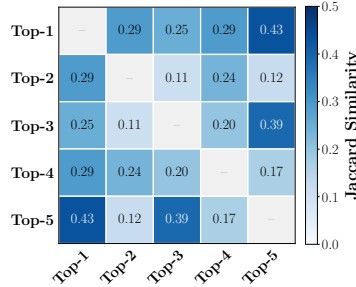

*(a)* Top-3 learned graph topologies.

*(b)* Pairwise Jaccard similarity of top-5 graphs.

*Figure 3.* Analysis of learned topologies on MMLU-Redux. (a) Visualization of the top-3 most frequently selected graph structures with their density. (b) Pairwise Jaccard similarity between top-5 graphs, showing low structural overlap.

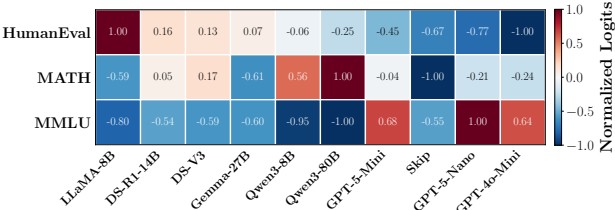

*Figure 4.* Dataset-level LLM selection preferences learned by **HieraMAS**. Normalized Logits indicate selection preference, with higher values indicating stronger preference.

*Table 4.* Cost comparison ($). Lowest cost in **bold**.

| Method | HumanEval++ | MATH | MMLU |
|---|---|---|---|
| GDesigner | 1.33 | 2.16 | 3.66 |
| MASRouter | **0.12** | **0.32** | **0.15** |
| **Ours** | 0.53 | 1.52 | 1.29 |

Router and GDesigner across the three benchmarks, as shown in Table 4. MASRouter achieves the lowest cost, which can be attributed to two factors: (1) its reward function explicitly incorporates cost penalties, and (2) following their original implementation, we constrain the maximum number of agents to 6, resulting in aggressive role pruning. In contrast, **HieraMAS** with SKIP action does not impose such hard constraints; instead, it learns to reduce costs through sparser communication topologies rather than pruning agents, achieving a better trade-off between cost and performance. Compared to GDesigner, which incurs the highest cost, the difference stems from its lack of LLM and role selection: it defaults to using the strongest (and most expensive) model for all agents. Furthermore, GDesigner requires 3 rounds of agent communication, whereas **HieraMAS** completes inference in a single round, further contributing to the cost reduction.

## 5. Related Work

**Multi-Agent Systems.** LLM-based multi-agent systems have emerged as a powerful paradigm, with foundational frameworks enabling agent collaboration through conversation programming, standardized operating procedures, and dynamic composition (Li et al., 2023; Hong et al., 2024; Qian et al., 2024; Chen et al., 2024; Niu et al., 2025; Yi et al., 2025; Shen et al., 2025). More recently, Mixture-of-Agents (Wang et al., 2024) introduces a layered architecture exploiting the collaborativeness phenomenon where LLMs generate better responses when provided with outputs from other models. Our work builds upon this insight by incorporating intra-node LLM mixtures within a broader MAS framework, while jointly optimizing communication topology and role configurations.

**Optimizing Multi-Agent Systems.** Recent research has focused on optimizing MAS from multiple perspectives: communication topology optimization via graph-based representations and reinforcement learning (Zhuge et al., 2024; Zhang et al., 2024a; 2025; Qian et al., 2025b), dynamic role assignment with agent importance scoring (Liu et al., 2023b), and LLM routing for cost-quality tradeoffs (Ong et al., 2024; Yue et al., 2025). Beyond these directions, AgentVerse (Chen et al., 2024) explores dynamic agent recruitment that adaptively assembles agent teams based on task complexity, while MetaGPT (Hong et al., 2024) introduces structured communication protocols inspired by software engineering workflows to reduce redundant interactions. While most of these works optimize individual dimensions or focus on single-agent routing, **HieraMAS** jointly optimizes topology and intra-node LLM configurations for multi-agent collaboration within a unified framework.

## 6. Conclusion

We presented **HieraMAS**, a framework that unifies intra-node and inter-node collaboration in multi-agent systems. By introducing supernodes with internal LLM mixtures

and a two-stage training algorithm, `HieraMAS` addresses the fundamental credit assignment challenges: multi-level rewards resolve per-node attribution, while holistic graph classification circumvents intractable per-edge credit assignment. Experiments on diverse domains demonstrate that `HieraMAS` achieves state-of-the-art performance with superior cost-efficiency.

## Impact Statement

This paper presents work on MAS, which are increasingly prevalent across various application domains. We believe this research contributes positively to the broader field of machine learning and collaborative AI systems. We do not foresee any direct negative societal consequences arising from this work.

## Acknowledgements

This work is supported by the MBZUAI-WIS Joint Program for Artificial Intelligence Research.

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

# Appendix

## A. Notation Table

Table 5 summarizes the key notation used throughout this paper.

*Table 5.* Summary of notation used in this paper.

| Symbol | Description | Symbol | Description |
|---|---|---|---|
| *Search Space & Supernode Structure* | | | |
| $\mathbb{S}$ | Search space $(\mathbb{M}, \mathbb{R}, \mathbb{G})$ | $S_i$ | Supernode $i$ with role $r_i$ |
| $\mathbb{M}$ | Pool of $N_m$ LLM backbones | $r_i$ | Role for supernode $i$, $r_i \in \mathbb{R}$ |
| $\mathbb{R}$ | Set of $N_r$ predefined agent roles | $W$ | Number of proposers per supernode |
| $\mathbb{G}$ | Space of graph topologies | $m_{i,j}^{(w)}$ | LLM for $j$-th proposer in supernode $i$ |
| $\mathcal{G}$ | Pool of $K$ candidate DAGs | $m_i^{(a)}$ | LLM for synthesizer in supernode $i$ |
| $N$ | Max number of supernodes | | |
| *MDP Components & Policy* | | | |
| $\mathcal{X}$ | State space | $\pi_m$ | LLM selection policy |
| $\mathcal{A}$ | Action space | $\pi_\theta$ | Overall policy with parameters $\theta$ |
| $P$ | Transition function | $f_G$ | Graph classifier |
| $R$ | Reward function | $\tau$ | Softmax temperature |
| $\mathcal{Q}$ | Input query | $\mathbf{h}_{\mathcal{Q},r}$ | Query-role embedding |
| $a^*$ | Ground-truth answer | $\mathbf{h}_{m_\ell}$ | LLM profile embedding |
| $\mathbf{E}$ | Adjacency matrix $\in \{0,1\}^{N \times N}$ | | |
| *Rewards* | | | |
| $R^{\text{final}}$ | Final task reward | $U(\cdot)$ | Utility (correctness) function |
| $R_i^{\text{node}}$ | Node-level reward for supernode $i$ | $C(\cdot)$ | Cost function (token expenditure) |
| $R_i^{\text{eff}}$ | Effective reward (node + final) | $\lambda$ | Cost sensitivity hyperparameter |
| $\alpha$ | Mixing coefficient for multi-level reward | | |
| *Training Objectives* | | | |
| $\mathcal{L}_{\text{stage1}}$ | Stage 1 loss (LLM selector) | $T_1, T_2$ | Training iterations for Stage 1, 2 |
| $\mathcal{L}_{\text{stage2}}$ | Stage 2 loss (graph classifier) | $\mathcal{D}$ | Training dataset |
| $\lambda_H$ | Entropy regularization coefficient | $\mathcal{D}_G$ | Graph classification dataset |
| $H(\pi_m^{(p)})$ | Entropy at position $p$ | | |
| *Graph Classifier* | | | |
| $\mathbf{A}_k$ | Adjacency matrix for graph $G_k$ | $\mathbf{z}_G$ | Graph-level embedding |
| $\mathbf{X}$ | Node feature matrix | $s_k$ | Suitability score for $G_k$ |
| $\mathbf{Z}$ | Graph-aware node representations | $G^*$ | Selected optimal graph |

## B. Algorithm

Algorithm 1 presents the complete two-stage training procedure for **HieraMAS**.

## C. Theoretical Analysis

In this section, we provide formal theoretical analysis justifying our two-stage training design, corresponding to Theorem 3.1 in Section 3.4. The theorem identifies two fundamental credit assignment challenges in multi-agent systems: (i) per-node credit assignment, where individual agent failures may be masked by system-level success, and (ii) per-edge credit assignment, where the contribution of individual communication links is difficult to isolate. Below, we formalize these challenges and prove how our proposed solutions: multi-level rewards and holistic graph selection to address them.

**Problem Setting.** We consider multi-agent coordination problems formalized as MDPs with bounded rewards $R \in [-B, B]$, finite state space $\mathcal{X}$, and finite action space $\mathcal{A}$ (corresponding to discrete LLM and topology selections). The communication topology is represented as a directed acyclic graph (DAG) over $N$ supernodes.

---

**Algorithm 1 HieraMAS Two-Stage Training**

---

**Require:** Training set $\mathcal{D}$, graph pool $\mathcal{G}$, LLM pool $\mathbb{M}$
**Ensure:** Trained LLM selector $\pi_m$, graph classifier $f_G$

    **Stage 1: Supernode Optimization**
1: **for** iteration $= 1, \ldots, T_1$ **do**
2:     Sample batch $\{\mathcal{Q}_b\}_{b=1}^{B}$ from $\mathcal{D}$
3:     **for** each query $\mathcal{Q}$ in batch **do**
4:         Sample random graph $G_k \sim \text{Uniform}(\mathcal{G})$
5:         Sample LLM assignments $\mathbf{m} \sim \pi_m(\cdot|\mathcal{Q})$
6:         Execute MAS with topology $G_k$ and LLMs $\mathbf{m}$, obtain output $\hat{a}$
7:         Compute rewards using ground-truth: $\{R_i^{\text{node}}\}$ and $R^{\text{final}}$
8:         Compute effective rewards $R_i^{\text{eff}}$ via Eq. equation 7
9:     **end for**
10:    Update $\pi_m$ by minimizing $\mathcal{L}_{\text{stage1}}$ (Eq. equation 9)
11: **end for**

    **Stage 2: Graph Classifier Training**
12: Freeze LLM selector $\pi_m$
13: $\mathcal{D}_G \leftarrow \emptyset$                                                  ▷ Graph classification dataset
14: **for** each query $\mathcal{Q}$ in $\mathcal{D}$ **do**
15:     **for** $k = 1, \ldots, M$ **do**
16:         Sample graph $G_k$ from $\mathcal{G}$, execute MAS, compute reward $R_k$
17:     **end for**
18:     Label top graphs with $R_k > 0$ as positive; add to $\mathcal{D}_G$
19: **end for**
20: **for** iteration $= 1, \ldots, T_2$ **do**
21:    Update graph classifier $f_G$ on $\mathcal{D}_G$ via Eq. equation 10
22: **end for**

---

Section C.1 provides the formal proof for Theorem 3.1(i), showing how multi-level rewards guarantee correct gradient direction for all supernodes by preventing the masking effect. Section C.2 proves Theorem 3.1(ii), demonstrating that per-edge optimization incurs irreducible credit assignment error, while our holistic graph selection approach reduces this to a vanishing estimation error.

### C.1. Per-Node Credit Assignment: Multi-Level Rewards

This subsection provides the formal proof for Theorem 3.1(i), which addresses the per-node credit assignment challenge. As illustrated in Figure 1(a), when using only final task rewards, a failing supernode (e.g., a Reviewer that introduces errors) may receive positive gradient updates if other agents compensate for its mistakes, leading to incorrect policy reinforcement. Our multi-level reward mechanism (Eq. equation 7) addresses this by combining node-level rewards $R_i^{\text{node}}$ with the final reward $R^{\text{final}}$, ensuring that each supernode receives feedback proportional to its individual contribution.

We provide theoretical justification for why multi-level rewards lead to more accurate policy gradients compared to using final rewards alone. The key insight is that when agents have heterogeneous performance, final-reward-only training can push poorly-performing agents' policies in incorrect directions.

**Proposition 1** (Gradient Bias under Final Reward). *Consider a multi-agent system with $N$ supernodes, where the policy for selecting LLM configuration at supernode $i$ is parameterized by $\theta_i$. Let $R^{final}$ denote the final task reward and $R_i^{node}$ denote the node-level reward for supernode $i$. Define the policy gradient under final reward as:*

$$g_i^{final} = \nabla_{\theta_i} \log \pi_{\theta_i}(m_i|\mathcal{Q}) \cdot R^{final}, \tag{11}$$

*and the policy gradient under multi-level reward as:*

$$g_i^{multi} = \nabla_{\theta_i} \log \pi_{\theta_i}(m_i|\mathcal{Q}) \cdot R_i^{eff}, \tag{12}$$

where $R_i^{eff} = \alpha \cdot R_i^{node} + (1 - \alpha) \cdot R^{final}$.

Suppose there exists a "failed" supernode $j$ such that $R_j^{node} < 0$, while the final reward is positive due to other agents compensating for this failure, i.e., $R^{final} > 0$. Then:

(i) Under final reward, the gradient $g_j^{final}$ reinforces the current (suboptimal) policy for supernode $j$.

(ii) Under multi-level reward with $\alpha > \frac{R^{final}}{R^{final} - R_j^{node}}$, the gradient $g_j^{multi}$ correctly penalizes the suboptimal policy.

*Proof.* We analyze the direction of policy updates for supernode $j$.

**Part (i): Final Reward Case.** The policy gradient update rule increases the probability of actions that receive positive reward and decreases the probability for actions with negative reward. Under the REINFORCE estimator (Williams, 1992), the expected update direction for supernode $j$ is:

$$\mathbb{E}\left[g_j^{\text{final}}\right] = \mathbb{E}\left[\nabla_{\theta_j} \log \pi_{\theta_j}(m_j|\mathcal{Q}) \cdot R^{\text{final}}\right]. \tag{13}$$

Since $R^{\text{final}} > 0$ by assumption, this gradient will increase the log-probability of the sampled action $m_j$. However, supernode $j$ produced a suboptimal output (as evidenced by $R_j^{\text{node}} < 0$). Therefore, the policy is reinforced in a direction that maintains or increases the probability of selecting the suboptimal configuration, the gradient direction is incorrect with respect to improving supernode $j$'s individual performance.

**Part (ii): Multi-Level Reward Case.** Under the multi-level reward, the effective reward for supernode $j$ is:

$$R_j^{\text{eff}} = \alpha \cdot R_j^{\text{node}} + (1 - \alpha) \cdot R^{\text{final}}. \tag{14}$$

For the gradient to correctly penalize the suboptimal policy, we require $R_j^{\text{eff}} < 0$, which implies:

$$\alpha \cdot R_j^{\text{node}} + (1 - \alpha) \cdot R^{\text{final}} < 0 \tag{15}$$

$$(1 - \alpha) \cdot R^{\text{final}} < -\alpha \cdot R_j^{\text{node}} \tag{16}$$

$$(1 - \alpha) \cdot R^{\text{final}} < \alpha \cdot |R_j^{\text{node}}| \quad \text{(since } R_j^{\text{node}} < 0) \tag{17}$$

Rearranging:

$$R^{\text{final}} - \alpha \cdot R^{\text{final}} < \alpha \cdot |R_j^{\text{node}}| \tag{18}$$

$$R^{\text{final}} < \alpha \cdot (R^{\text{final}} + |R_j^{\text{node}}|) \tag{19}$$

$$R^{\text{final}} < \alpha \cdot (R^{\text{final}} - R_j^{\text{node}}) \tag{20}$$

$$\alpha > \frac{R^{\text{final}}}{R^{\text{final}} - R_j^{\text{node}}}. \tag{21}$$

When this condition holds, $R_j^{\text{eff}} < 0$, and the gradient $g_j^{\text{multi}}$ will decrease the probability of the suboptimal action, correctly updating the policy in the direction that discourages the failing configuration. $\square$

**Corollary C.1** (Sufficient Condition for Gradient with Correct Sign). *If the reward signals are normalized such that $R^{final}, R_i^{node} \in [-1, 1]$, then setting $\alpha \geq 0.5$ ensures the gradient sign matches the desired update direction whenever $R_j^{node} \leq 0$ and $R^{final} \leq 1$.*

*Proof.* The worst case occurs when $R^{\text{final}} = 1$ and $R_j^{\text{node}} = 0$ (marginal failure). The threshold becomes $\alpha^* = \frac{1}{1-0} = 1$. However, for any strictly negative $R_j^{\text{node}} < 0$, we have:

$$\alpha^* = \frac{R^{\text{final}}}{R^{\text{final}} - R_j^{\text{node}}} < \frac{R^{\text{final}}}{R^{\text{final}}} = 1. \tag{22}$$

More practically, when $R_j^{\text{node}} = -1$ (complete failure) and $R^{\text{final}} = 1$:

$$\alpha^* = \frac{1}{1 - (-1)} = \frac{1}{2} = 0.5. \tag{23}$$

Thus, $\alpha \geq 0.5$ ensures the gradient sign matches the desired update direction for all cases where the node-level performance is at least as negative as the final reward is positive, in magnitude. $\square$

This analysis provides theoretical grounding for our multi-level reward design in Eq. equation 7. By incorporating node-level feedback, **HieraMAS** avoids the pathological case where high-performing agents mask the failures of others, enabling more accurate credit assignment and faster convergence to optimal supernode configurations.

### C.2. Per-Edge Credit Assignment: Holistic Graph Selection

This subsection provides the formal proof for Theorem 3.1(ii), which addresses the per-edge credit assignment challenge. As illustrated in Figure 1(b), when optimizing communication topology via per-edge policy gradients, individual edges receive credit based on the final task reward. However, after Stage 1 optimization, the LLM selector is already well-trained, causing most configurations to achieve positive rewards regardless of topology. This leads to an *irreducible* credit assignment error: non-beneficial edges are incorrectly reinforced simply because they co-occur with successful task completions.

Our holistic graph selection approach (Section 3.2) circumvents this problem by treating topology selection as a graph classification problem rather than per-edge optimization. By evaluating entire topologies as indivisible units and selecting from a pre-generated candidate pool $\mathcal{G}$, we transform the ill-posed per-edge credit assignment into a well-posed estimation problem with vanishing error.

We now analyze the credit assignment problem in per-edge topology optimization and justify our holistic graph selection approach.

**Proposition 2** (Credit Assignment Error in Per-Edge Optimization). *Consider optimizing graph topology via per-edge policy gradient after the LLM mixture has been optimized (Stage 1 converged). Let $G^* \subseteq \mathcal{E}$ denote the set of edges in the optimal topology, where $\mathcal{E}$ is the set of all possible edges. For each edge $e \in \mathcal{E}$, let $\pi_\phi(e)$ denote the probability of including edge $e$, parameterized by $\phi$. The per-edge policy gradient is:*

$$g_e = \nabla_\phi \log \pi_\phi(e) \cdot R^{\text{final}}, \quad \text{if } e \text{ is sampled.} \tag{24}$$

*Assumptions:*

**(A1)** *After Stage 1 optimization, the final reward is high with probability $q > 0.5$, i.e., $\Pr(R^{\text{final}} > 0) = q$.*

**(A2)** *Each edge $e$ is sampled independently with probability $p = 0.5$.*

*Justification of Assumptions:*

**(A1)** *Stage 1 optimizes the LLM mixture over the expectation of randomly sampled graphs. We posit that the capability of the LLM combination has a larger impact on task success than the communication topology, i.e., strong LLMs can compensate for suboptimal topologies, but weak LLMs cannot be saved by optimal topologies. Therefore, after Stage 1 converges, the optimized LLM selector achieves positive reward on more than half of the instances, i.e., $q > 0.5$.*

**(A2)** *Since we use randomly generated graphs during Stage 1 training, edges are sampled uniformly at random. For simplicity, we assume $p = 0.5$, corresponding to a uniform edge inclusion probability.*

*Let $\mathcal{E}^+ = G^*$ denote the set of beneficial edges (those in the optimal topology) and $\mathcal{E}^- = \mathcal{E} \setminus G^*$ denote the set of non-beneficial edges. Define the* gradient direction error *for an edge $e$ as:*

$$Error(e) = \begin{cases} 1 & \text{if } e \in \mathcal{E}^+ \text{ and } g_e < 0 \text{ (beneficial edge incorrectly weakened)} \\ 1 & \text{if } e \in \mathcal{E}^- \text{ and } g_e > 0 \text{ (non-beneficial edge incorrectly reinforced)} \\ 0 & \text{otherwise (correct gradient direction)} \end{cases} \tag{25}$$

*Then the expected fraction of edges with incorrect gradient direction is:*

$$\mathbb{E}\left[\frac{1}{|\mathcal{E}|}\sum_{e\in\mathcal{E}} Error(e)\right] \geq (1-\rho)\cdot p\cdot q, \tag{26}$$

*where $\rho = \frac{|G^*|}{|\mathcal{E}|}$ is the fraction of edges in the optimal topology.*

*Proof.* We analyze the gradient direction for edges in $\mathcal{E}^+$ and $\mathcal{E}^-$ separately.

**Case 1: Non-beneficial edges ($e \in \mathcal{E}^-$).**

Consider an edge $e \in \mathcal{E}^-$ that is sampled (included in the current graph $G$). The gradient update is:

$$g_e = \nabla_\phi \log \pi_\phi(e) \cdot R^{\text{final}}. \tag{27}$$

Since $\nabla_\phi \log \pi_\phi(e)$ points in the direction that increases $\pi_\phi(e)$, and $R^{\text{final}} > 0$ with probability $q$ (by Assumption 2), the gradient $g_e > 0$ will *increase* the probability of selecting edge $e$.

However, $e \notin G^*$, meaning this edge is not part of the optimal topology. Therefore, reinforcing $e$ is incorrect, the gradient pushes the policy toward including a suboptimal edge.

The probability that a non-beneficial edge $e \in \mathcal{E}^-$ receives an incorrect gradient update is:

$$\Pr(Error(e) = 1 \mid e \in \mathcal{E}^-) = \Pr(e \text{ sampled}) \cdot \Pr(R^{\text{final}} > 0) = p \cdot q. \tag{28}$$

**Case 2: Beneficial edges ($e \in \mathcal{E}^+$).**

For edges $e \in \mathcal{E}^+ = G^*$, correct gradient direction requires $g_e > 0$ (reinforcement). When $R^{\text{final}} > 0$, sampled beneficial edges receive correct positive gradients. However, when $R^{\text{final}} < 0$ (which occurs with probability $1 - q$), beneficial edges that were sampled receive incorrect negative gradients.

The probability that a beneficial edge receives an incorrect gradient:

$$\Pr(Error(e) = 1 \mid e \in \mathcal{E}^+) = \Pr(e \text{ sampled}) \cdot \Pr(R^{\text{final}} < 0) = p \cdot (1-q). \tag{29}$$

**Total Expected Error.**

The expected error across all edges is:

$$\mathbb{E}\left[\sum_{e\in\mathcal{E}} Error(e)\right] = \sum_{e\in\mathcal{E}^-} \Pr(Error(e) = 1) + \sum_{e\in\mathcal{E}^+} \Pr(Error(e) = 1) \tag{30}$$

$$= |\mathcal{E}^-| \cdot p \cdot q + |\mathcal{E}^+| \cdot p \cdot (1-q) \tag{31}$$

$$= p \cdot [(|\mathcal{E}| - |G^*|) \cdot q + |G^*| \cdot (1-q)] \tag{32}$$

$$= p \cdot [|\mathcal{E}| \cdot q - |G^*| \cdot q + |G^*| - |G^*| \cdot q] \tag{33}$$

$$= p \cdot [|\mathcal{E}| \cdot q + |G^*| \cdot (1-2q)] . \tag{34}$$

Dividing by $|\mathcal{E}|$ and letting $\rho = \frac{|G^*|}{|\mathcal{E}|}$:

$$\mathbb{E}\left[\frac{1}{|\mathcal{E}|}\sum_{e\in\mathcal{E}} Error(e)\right] = p \cdot [q + \rho \cdot (1-2q)] \tag{35}$$

$$= p \cdot [q + \rho - 2\rho q] \tag{36}$$

$$= p \cdot [q(1 - 2\rho) + \rho] . \tag{37}$$

When $q$ is high (close to 1) and $\rho$ is small (sparse optimal topology), the dominant term is:

$$\mathbb{E}[\text{Error Rate}] \approx p \cdot q \cdot (1 - 2\rho) + p \cdot \rho \approx p \cdot q \cdot (1 - \rho), \tag{38}$$

for $\rho \ll 1$. Since $p \cdot q$ represents the probability of sampling an edge under high reward, and $(1 - \rho)$ is the fraction of non-beneficial edges, this shows that:

$$\mathbb{E}[\text{Error Rate}] \geq (1 - \rho) \cdot p \cdot q. \tag{39}$$

$\square$

The error rate $(1 - \rho) \cdot p \cdot q$ reveals a fundamental tension in per-edge optimization after Stage 1. With $p = 0.5$ and $q > 0.5$, the error rate exceeds $(1 - \rho)/4$. The error is amplified when the optimal topology is sparse ($\rho \to 0$) or when Stage 1 achieves high success rate.

**Corollary C.2** (Justification for Holistic Graph Selection). *The high error rate in per-edge optimization motivates treating topology selection as a holistic graph classification problem (Section 3.2). By evaluating entire topologies as indivisible units and selecting from a pre-generated candidate pool, our approach avoids the per-edge credit assignment problem entirely.*

### C.3. Generalization Guarantee for Graph Classifier

We now provide theoretical guarantees for our graph classifier approach.

**Theorem C.3** (Generalization Guarantee for Graph Classifier). *Let $\mathcal{G} = \{G_1, G_2, \ldots, G_K\}$ be the pre-generated graph candidate pool, and let $\pi_m^*$ denote the optimized LLM selector from Stage 1.*

*Assumptions:*

**(B1)** *(Optimal topology coverage) The candidate pool $\mathcal{G}$ contains an optimal graph $G^* \in \mathcal{G}$.*

**(B2)** *(Reward separability) Let $\mu_k = \mathbb{E}[R(\mathcal{Q}, G_k)]$ denote the expected reward for graph $G_k$. There exists a margin $\gamma > 0$ such that:*

$$\mu^* - \mu_k \geq \gamma, \quad \forall G_k \neq G^*. \tag{40}$$

**(B3)** *(Bounded reward) The reward is bounded: $R(\mathcal{Q}, G_k) \in [-B, B]$ for all $\mathcal{Q}, G_k$.*

*Justification of Assumptions:*

**(B1)** *In practical settings, the number of roles is typically small. For instance, MATH uses 3 roles and MMLU-Redux uses 8 predefined roles at maximum, resulting in a finite and tractable search space of candidate topologies. Therefore the probability of including the optimal or near-optimal topology is high. Furthermore, if the exact optimal graph is not in $\mathcal{G}$, a near-optimal graph with similar structure is likely to be present.*

**(B2)** *This assumption is necessary and reflects our belief that the optimal communication topology leads to better performance given the same supernode configuration. If all topologies performed identically, topology optimization would be meaningless. The margin $\gamma$ captures the performance gap between optimal and suboptimal topologies.*

**(B3)** *This assumption is directly satisfied by our reward formulation in Eq. 8, where the utility and normalized cost ensure bounded rewards.*

*In Stage 2, for each graph $G_k$, we collect $N_k$ i.i.d. samples and compute the empirical mean reward:*

$$\hat{\mu}_k = \frac{1}{N_k} \sum_{j=1}^{N_k} R(\mathcal{Q}_j, G_k). \tag{41}$$

*Then, if $N_k \geq \frac{8B^2 \log(2K/\delta)}{\gamma^2}$ for all $k \in [K]$, with probability at least $1 - \delta$:*

$$\arg\max_{G_k \in \mathcal{G}} \hat{\mu}_k = G^*. \tag{42}$$

*Proof.* By Hoeffding's inequality, for bounded random variables in $[-B, B]$:

$$\Pr\left[|\hat{\mu}_k - \mu_k| > \epsilon\right] \leq 2\exp\left(-\frac{2N_k\epsilon^2}{(2B)^2}\right) = 2\exp\left(-\frac{N_k\epsilon^2}{2B^2}\right). \tag{43}$$

Applying union bound over all $K$ graphs:

$$\Pr\left[\exists k : |\hat{\mu}_k - \mu_k| > \epsilon\right] \leq 2K\exp\left(-\frac{N_k\epsilon^2}{2B^2}\right). \tag{44}$$

Setting $\epsilon = \gamma/4$ and $N_k \geq \frac{8B^2\log(2K/\delta)}{\gamma^2}$:

$$2K\exp\left(-\frac{1}{2B^2} \cdot \frac{8B^2\log(2K/\delta)}{\gamma^2} \cdot \frac{\gamma^2}{16}\right) = 2K\exp(-\log(2K/\delta)) = \delta. \tag{45}$$

Thus, with probability at least $1 - \delta$, for all $k$ simultaneously: $|\hat{\mu}_k - \mu_k| \leq \gamma/4$.

Under this event, for any $G_k \neq G^*$:

$$\hat{\mu}^* - \hat{\mu}_k \geq (\mu^* - \frac{\gamma}{4}) - (\mu_k + \frac{\gamma}{4}) \tag{46}$$

$$= (\mu^* - \mu_k) - \frac{\gamma}{2} \tag{47}$$

$$\geq \gamma - \frac{\gamma}{2} = \frac{\gamma}{2} > 0. \tag{48}$$

Therefore, $G^* = \arg\max_{G_k} \hat{\mu}_k$. $\qquad\square$

This theorem reveals a fundamental difference from per-edge policy gradient:

- **Per-edge RL** (Proposition 2): The gradient error $\Omega((1-\rho)pq)$ is *irreducible*, and persists regardless of sample size due to inherent credit assignment ambiguity.

- **Graph classifier** (Theorem C.3): The estimation error $|\hat{\mu}_k - \mu_k| = \mathcal{O}(B/\sqrt{N_k})$ *vanishes* with more samples. With $N_k = \mathcal{O}(B^2\log(K)/\gamma^2)$, the optimal graph is identified with high probability.

The key insight is that: treating graph selection as supervised learning transforms an ill-posed credit assignment problem into a well-posed estimation problem.

## D. Additional Experimental Details

### D.1. Implementation Details

**Training Procedure.**   Our training procedure consists of two stages:

**Stage 1 (Supernode Optimization):** We optimize the LLM selector $\pi_m$ using the Adam optimizer (Kingma, 2014) with learning rate $2 \times 10^{-3}$ and weight decay $5 \times 10^{-4}$. During this stage, graph topologies are randomly sampled from the candidate pool $\mathcal{G}$ to expose the LLM selector to diverse communication patterns. The mixing coefficient $\alpha$ for multi-level rewards is set to 0.5 based on the theoretical analysis in Corollary C.1. The training iterations $T_1$ is searched over $\{5, 10\}$ with batch size 8.

**Stage 2 (Graph Classifier Training):** After Stage 1 converges, we freeze the LLM selector $\pi_m$ and train the GCN-based graph classifier. For each training instance, we sample $M = 5$ random graphs from the candidate pool. We select the top-2 highest reward graphs as positive samples and the remaining 3 as negative samples. The GCN encoder consists of 2 graph convolutional layers with hidden dimension 256, followed by mean pooling and a linear classifier. We tune the dropout rate across $\{0.05, 0.1, 0.2, 0.5\}$ and select the best model based on validation performance. We train for $T_2 = 20$ epochs with early stop.

*Table 6.* Effect of the number of proposers per supernode on MMLU-Redux.

| Method | Acc (%) | Cost ($) |
|---|---|---|
| w/ 8 proposers | 88.80 | 2.32 |
| w/ 6 proposers | 91.20 | 1.97 |
| **Ours** (2 proposers) | **95.20** | **1.29** |

*Table 7.* Effect of graph pool size $K$ on MMLU-Redux.

| Method | Acc (%) |
|---|---|
| w/ $K = 50$ | 93.60 |
| w/ $K = 500$ | 94.40 |
| **Ours** ($K = 200$) | **95.60** |

**Graph Candidate Pool.** We maintain a candidate pool of $K = 200$ randomly generated directed acyclic graphs (DAGs) for both training and inference. The graphs are generated with varying edge densities sampled uniformly from $[0.3, 0.75]$ to ensure diversity, covering sparse, medium, and dense connectivity patterns.

**Sentence Encoder.** For query and role encoding, we use the pre-trained `all-MiniLM-L6-v2` sentence transformer (Reimers & Gurevych, 2019) to compute embeddings $\mathbf{h}_{\mathcal{Q},r}$ and $\mathbf{h}_{m_\ell}$. The encoder produces 384-dimensional embeddings. The model is frozen without finetuning.

**Data Splits.** For all benchmarks, we use an 80%/20% train/validation split. The training set is used for both Stage 1 and Stage 2 optimization, while the validation set is used for model selection (dropout tuning) and early stopping. Final evaluation is performed on the official test sets of each benchmark.

### D.2. Effect of Proposers in Supernode

We investigate the impact of the number of proposers per supernode on MMLU-Redux, as shown in Table 6. Increasing the number of proposers from 2 to 6 or 8 leads to significant performance degradation: accuracy drops from 95.20% (2 proposers) to 91.20% (6 proposers) and 88.80% (8 proposers).

Interestingly, the cost does not increase proportionally with the number of proposers. We observe that during training, the system increasingly selects the `skip` action for synthesizer positions as the number of proposers grows, effectively pruning some roles from the MAS. This adaptive behavior differs from the 2-proposer configuration where pruning is less frequent.

The performance drop with more proposers can be attributed to two factors. First, when the system learns to handle the increased complexity of many proposers, some meaningful roles may be inadvertently pruned during the optimization process, reducing the diversity of perspectives available for synthesis. Second, when too many proposers are active, the synthesizer receives excessively long context from multiple proposer outputs, leading to the *lost-in-the-middle* phenomenon (Liu et al., 2024) where critical information is overlooked. These findings suggest that a moderate number of proposers (e.g., 2) strikes the optimal balance between diversity of perspectives and the synthesizer's aggregation capacity.

### D.3. Effect of Graph Pool Size

We investigate the impact of the graph candidate pool size $K$ on MMLU-Redux, as shown in Table 7. The results demonstrate that **HieraMAS** is not sensitive to the choice of $K$: performance remains stable across different pool sizes, with accuracy ranging from 93.60% ($K = 50$) to 95.60% ($K = 200$).

### D.4. Case Studies

We present qualitative case studies from each benchmark to illustrate how **HieraMAS** adaptively selects LLM configurations and communication topologies based on task characteristics.

*Table 8.* Case studies on MMLU-Redux. The typical configuration assigns GPT-5-Mini or GPT-5-Nano as proposers and GPT-5-Nano as the aggregator, favoring cost-efficient models for knowledge-intensive QA tasks.

| MMLU-Redux Case Studies | |
|---|---|
| **Case 1: Nutrition Question** | **Case 2: Communication Theory** |
| **Query:** Which of the following appears to lower bad cholesterol?
A. Vitamin D     B. Niacin
C. Thiamine     D. Riboflavin | **Query:** According to "measurement," what is the step that occurs between an individual gaining information and changing behavior?
A. coorientation     B. opinion change
C. reaction formation     D. semantic encoding |

## D.5. Training Cost Analysis

Table 11 presents the training cost comparison across different learning-based methods. AFlow incurs substantially higher training costs due to its Monte Carlo Tree Search optimization with Claude 3.5-Sonnet. Our method is more cost-efficient than GDesigner, which lacks LLM selection and defaults to using the strongest model (GPT-5-Mini) for all agents, whereas our approach incorporates LLM selection with cost-aware rewards that adaptively choose cheaper models when appropriate. Compared to MASRouter, our training cost is slightly higher due to two factors: (1) different LLM selection preferences during training, and (2) our two-stage algorithm requires additional sample collection in Stage 2 for graph classifier training, which incurs extra cost.

*Table 11.* Training cost comparison (in USD) across benchmarks.

| Dataset | GDesigner | AFlow | MASRouter | Ours |
|---|---|---|---|---|
| MATH | $1.22 | $18.62 | $1.97 | $2.03 |
| HumanEval++ | $1.95 | $28.64 | $0.54 | $1.55 |
| MMLU-Redux | $3.51 | $13.94 | $0.46 | $1.85 |

## D.6. Prompts

We provide the text profiles used for embedding-based LLM selection and role assignment. These profiles are encoded using a sentence transformer to compute similarity scores during the selection process.

**LLM Profiles.** Table 12 presents the text descriptions for each LLM option in our candidate pool. These profiles capture model capabilities, cost characteristics, and recommended use cases.

*Table 9.* Case studies on HumanEval. The typical configuration assigns Qwen3-8B, Qwen3-80B, or LLaMA3.1-8B as proposers, leveraging diverse code-specialized models for programming tasks.

| HumanEval Case Studies |
|---|

| **Case 1: Sum and Product** | **Case 2: Filter by Substring** |
|---|---|
| **Query:** `def sum_product(numbers: List[int]) -> Tuple[int, int]:` `""" For a given list of integers, return a tuple consisting of a sum and a product of all the integers in a list..."""` | **Query:** `def filter_by_substring(strings: List[str], substring: str) -> List[str]:` `""" Filter an input list of strings only for ones that contain given substring..."""` |

**Role Profiles.** Table 13 presents the text descriptions for each agent role. These profiles define role responsibilities and specializations used in the supernode configuration.

*Table 10.* Case studies on MATH. The typical configuration assigns Qwen3-8B combined with Qwen3-80B or DeepSeek-V3.2 as proposers, utilizing strong mathematical reasoning models.

| MATH Case Studies | |
|---|---|
| **Case 1: Geometry Reflection** | **Case 2: Cone Volume** |
| **Query:** Triangle $ABC$ with vertices $A(-2,0)$, $B(1,4)$ and $C(-3,2)$ is reflected over the $y$-axis to form triangle $A'B'C'$. What is the length of a segment drawn from $C$ to $C'$? | **Query:** The diameter of a cone is 30 decimeters. If the height is two times the radius, what is the volume of the cone in cubic decimeters? |

*Table 12.* LLM profiles used for embedding-based model selection. Each profile describes model strengths, cost tier, and recommended use cases.

| Model | Profile Description |
|---|---|
| qwen3-8b | Fast and efficient 8B parameter model from Alibaba. Very fast inference, low latency, cost-effective. Good for basic text generation and simple Q&A. Very low cost. |
| qwen3-80b | Powerful 80B mixture-of-experts model from Alibaba. Strong reasoning, excellent at complex tasks, good at math and coding. Good for complex reasoning and mathematical problem-solving. Medium-high cost. |
| deepseek-r1-14b | Math-focused 14B model optimized for reasoning. Excellent mathematical reasoning and step-by-step problem solving. Good for math problems and logical reasoning. Low-medium cost. |
| llama-8b | General-purpose 8B model from Meta. Well-balanced capabilities with good instruction following. Good for general tasks, conversation, and basic reasoning. Very low cost. |
| deepseek-v3 | State-of-the-art reasoning model with excellent performance. Top-tier reasoning, excellent at complex problems, strong coding ability. Good for complex reasoning and advanced math. Medium cost. |
| gemma-3-27b | Balanced 27B model from Google. Good balance of speed and capability with strong instruction following. Good for medium complexity tasks. Medium cost. |
| gpt-5-nano | Extremely fast and cheap model from OpenAI. Very low latency and lowest cost. Good for simple text processing and basic Q&A. Lowest cost option. |
| GPT-5-Mini | Capable model from OpenAI with strong reasoning. Strong reasoning and excellent at complex tasks. Good for complex reasoning and multi-step problems. Medium-high cost. |
| gpt-4o-mini | Capable and reliable model from OpenAI. Strong reasoning, good at coding, reliable instruction following. Good for complex reasoning and code generation. Medium-high cost. |
| skip | Special token: Do not assign any LLM to this position. Removes this agent from the pipeline entirely. No API call made, no cost incurred. Use when position is not needed for the current task. |

*Table 13.* Role profiles used for embedding-based role assignment. Roles are categorized by their primary application domain.

| Role | Profile Description |
|------|--------------------|
| *Mathematical Reasoning Roles (MATH benchmark)* | |
| Mathematical Analyst | Specialized in breaking down complex math problems. Identifies mathematical structure, defines variables, and sets up equations. Best for initial problem understanding and mathematical modeling. |
| Math Solver | Specialized in executing mathematical computations. Solves equations, performs calculations, and verifies numerical results. Best for carrying out calculations and deriving numerical answers. |
| Inspector | Specialized in reviewing and validating solutions. Checks solution correctness, identifies errors, and verifies final answers. Best for final verification and error catching. |
| *Programming Roles (HumanEval benchmark)* | |
| Project Manager | Specialized in design patterns and code structure. Coordinates solution strategy and organizes workflow. Best for complex problems requiring structured multi-step approaches. |
| Algorithm Designer | Specialized in algorithm design and pseudocode. Designs algorithms, creates pseudocode, and plans computational steps. Best for problems requiring novel algorithms or optimization. |
| Test Analyst | Specialized in test cases and edge conditions. Identifies edge cases, designs test scenarios, and validates solutions. Best for ensuring solution robustness across all inputs. |
| Programming Expert | Specialized in writing executable code. Implements algorithms, writes clean code, and handles edge cases. Best for tasks requiring actual code implementation. |
| Bug Fixer | Specialized in debugging and code correction. Identifies bugs, fixes errors, and improves code robustness. Best for correcting issues in existing code. |
| *Knowledge QA Roles (MMLU-Redux benchmark)* | |
| Knowledgeable Expert | Specialized in domain knowledge and research. Provides domain expertise and searches for relevant information. Best for problems requiring specialized domain knowledge. |
| Critic | Specialized in critical analysis and feedback. Critiques solutions, identifies weaknesses, and suggests improvements. Best for improving solution quality through critical review. |
| Mathematician | For STEM questions. Handles mathematical reasoning, calculations, and formal logic. Best for math, physics, and engineering questions. |
| Psychologist | For behavioral and social questions. Analyzes human behavior, cognitive processes, and social dynamics. Best for psychology and sociology questions. |
| Historian | For historical and cultural questions. Provides historical context, analyzes events, and identifies patterns. Best for history and political science questions. |
| Doctor | For health and biology questions. Covers medical knowledge, biological processes, and health advice. Best for medicine and biology questions. |
| Lawyer | For legal and ethical questions. Handles legal reasoning, case analysis, and ethical frameworks. Best for law and ethics questions. |
| Economist | For economic and business questions. Covers economic analysis, market dynamics, and financial reasoning. Best for economics, business, and finance questions. |
| Programmer | For computer science questions. Covers algorithm design, coding concepts, and system architecture. Best for computer science and technology questions. |

