# OpenReview forum: "HieraMAS: Optimizing Intra-Node LLM Mixtures and Inter-Node Topology for Multi-Agent Systems"
_ICML.cc/2026/Conference — ICML 2026 regular_

### Official Review · Reviewer_yjbr · 2026-03-09

**Soundness:** 2
**Presentation:** 3
**Significance:** 3
**Originality:** 2
**Overall Recommendation:** 4
**Confidence:** 4

**Summary:**

The paper proposes HieraMAS, a hierarchical framework that jointly optimizes intra-node LLM mixtures and inter-node communication topologies to enhance multi-agent collaboration. To address the credit assignment challenges inherent in this dual optimization, the authors introduce a two-stage training strategy that first utilizes multi-level rewards on random graphs to robustly train node-level policies and subsequently formulates topology selection as a holistic graph classification task. This approach is rigorously benchmarked on reasoning and coding tasks to demonstrate superior performance and cost-efficiency compared to existing topology optimization methods.

**Compliance With Llm Reviewing Policy:**

Affirmed.

**Final Justification:**

The author provided detailed responses to my concerns, and I am willing to raise the score to 4.

**Key Questions For Authors:**

Q1: Could the authors provide training curves for both the policy learner ($\pi_m$) and graph classifier ($f_G$) to demonstrate convergence stability?

Q2: With a fixed candidate pool of only 200 random graphs, how is the optimality of the selected topology guaranteed against the vast search space of possible DAGs?

Q3: Does training the LLM policy on random graphs result in a generic average strategy that fails to specialize for specific structural roles such as hubs versus leaves in the optimal topology? Did you evaluate whether fine-tuning the policy after fixing the topology or optimizing both simultaneously yields superior performance compared to the proposed decoupled approach?

Q4: Given the performance degradation observed with increased proposers in Table 6, does the Supernode benefit derive from true multi-model mixture or merely from heterogeneous selection? Have you compared the method against a baseline where each node selects only one LLM ($W=1$) to isolate the specific gains of the propose-synthesis structure under equivalent token costs?

**Limitations:**

The limitations of the study should be stated more explicitly.

**Strengths And Weaknesses:**

# Soundness
This work anchors its methodology in a two-stage reinforcement learning framework, yet it fails to demonstrate the iterative performance dynamics required to validate optimization stability. Specifically, the manuscript lacks learning curves for the policy learner and graph classifier. Furthermore, the experimental design does not sufficiently disentangle whether performance gains derive from the novel Supernode architecture or merely from selecting superior individual models, preventing a definitive assessment of the complex structure's necessity compared to simpler selection-based baselines.

# Presentation
This work is clearly written and well-structured, effectively articulating the HieraMAS framework and decomposing the complex joint optimization problem into a tractable two-stage process involving Supernode optimization and Graph Classification. Furthermore, it situates itself within the literature by distinguishing its holistic topology selection from previous per-edge optimization methods, supported by intuitive visualizations such as the credit assignment challenges diagram that make the narrative easy to follow.

# Significance
This work addresses the critical challenge of jointly optimizing intra-node agent configurations and inter-node communication topologies, moving beyond existing approaches that treat these dimensions in isolation. By introducing the Supernode architecture and a decoupled two-stage learning strategy, it advances the field's capability to resolve complex credit assignment issues in multi-agent systems. This methodological shift provides substantial practical utility for developing cost-effective, high-performing agent teams that avoid the computational overhead of fully connected systems.

# Originality
This work presents a practical engineering framework that integrates existing techniques such as intra-node LLM selection and inter-node topology retrieval rather than introducing fundamental methodological innovations. The Supernode architecture adapts the established Mixture-of-Agents concept while the topology optimization is simplified from a generative problem to a classification task over a fixed pool. Consequently, the contribution represents an incremental system design instead of a novel theoretical or algorithmic advance.

---

> ### Author Rebuttal · Authors · 2026-03-31
>
> We are grateful to Reviewer yjbr for the thorough evaluation and recognition of our significance and presentation. We address each point below.
>
> **Q1: Convergence stability**
>
> We provide the training curves for both stages on MATH and MMLU-Redux from our training logs:
>
> - **Stage 1 (Policy Learner)**: https://anonymous.4open.science/r/hieramas-f350/stage1.pdf
> - **Stage 2 (Graph Classifier)**: https://anonymous.4open.science/r/hieramas-f350/stage2.pdf
>
> For Stage 1, both MATH and MMLU-Redux show rapid initial improvement in solve rate, while the mean reward continues to increase afterward. Since our reward (Eq. 8) jointly accounts for correctness and token cost, the continued mean reward improvement indicates that HieraMAS learns to **assign cheaper LLMs** to appropriate positions, achieving better cost-performance trade-offs. For Stage 2, both train and validation loss decrease steadily on MATH and MMLU-Redux, with no overfitting observed as validation loss tracks closely with training loss.
>
> **Q2: DAG search space**
>
> We acknowledge that the fixed pool size may pose a limitation in theory. However, in most practical scenarios, the number of roles is limited (e.g., 3 for programming, 5 for MATH), resulting in a manageable search space. Furthermore, inspired by recent findings that irregular and sparse topologies outperform regular ones[1,2], we prune graphs with density ratio > 0.7, which further reduces search space. Empirically, the top-ranked topologies (Figure 3a) have density ratios of **0.23–0.32**, confirming that optimal solutions concentrate in the sparse region.
>
> Moreover, by reformulating topology selection as graph classification, expanding the pool simply requires generating more annotated (graph, reward) training pairs in Stage 2, without modifying the pipeline.
>
> **Q3: Structure-aware specialization**
>
> We acknowledge that $\pi\_m(m\_\ell \mid Q, r)$ does not encode topology position. To empirically verify whether topology-aware selection would help, we augment the policy to $\pi\_m(m\_\ell \mid Q, r, s)$ on MMLU-Redux (GPT-5-Mini, 9 roles), where $s$ encodes the node's structural role in $G^*$ (in-degree, out-degree, is\_leaf with out-degree=0, is\_hub with in-degree$\geq$5):
>
> Table: Performance comparison with structure-aware HieraMAS on MMLU-Redux
> | Method | MMLU-Redux Acc (%) |
> |---|---|
> | AFlow | 91.20 |
> | GDesigner | 92.00 |
> | MASRouter | 88.33 |
> | **Ours** | **95.20** |
> | Ours (structure-aware) | 94.40 |
>
> The structure-aware variant slightly decreases performance, suggesting that structural information introduces noise rather than useful signal. This indicates that LLM selection is primarily driven by task characteristics and role requirements rather than topological position.
>
> **Q3 (cont'd): Decoupled vs. joint**
>
> We clarify that MASRouter serves as a baseline for joint optimization, as it learns LLM routing and topology selection jointly. As shown in Table 1, MASRouter underperforms HieraMAS across all benchmarks. Our theoretical analysis in Appendix C.2 further supports this observation: joint optimization suffers from per-edge credit assignment entanglement, where LLM selection biases can distort gradient signals for topology learning.
>
> **Q4: Multi-model mixture vs. heterogeneous selection**
>
> We appreciate this concern. We have conducted further analysis and find that increasing proposers introduces a **heterogeneity-accuracy trade-off**: more diverse proposals bring richer perspectives, but also more erroneous responses that mislead the synthesizer. Recent work [3] has similarly identified incorrect proposals as a primary cause of MAS performance degradation.
>
> We supplement with W=1, W=3, and W=8 with adaptive pruning (stopping when cumulative probability exceeds 90%):
>
> Table: Performance comparison with varying #proposers
> | Config | MATH (%) | MMLU-Redux (%) |
> |--------|----------|----------------|
> | W=1 | 95.56 | 91.20 |
> | **Ours (W=2)** | 96.67 | 95.20 |
> | W=3 | 98.89 | 95.20 |
> | W=6 | 91.11 | 91.20 |
> | W=8 | — | 88.80 |
> | W=8 (adaptive) | — | 92.00 |
>
> W=1 lacks heterogeneity and underperforms W=2 and W=3, confirming that multi-model mixture indeed helps. As W grows further (W=6, W=8), erroneous responses degrade performance. W=8 (adaptive) recovers from 88.80% to 92.00% by pruning inappropriate proposers, further validating the heterogeneity-accuracy trade-off.
>
> ---
>
> We hope the above responses address your concerns. We would be happy to provide further clarification if needed.
>
>
> **References**
>
> [1] Qian et al., Scaling Large Language Model-based Multi-Agent Collaboration, ICLR 2025.
>
> [2] Zhang et al., G-Designer: Architecting Multi-Agent Communication Topologies via Graph Neural Networks, ICML 2025
>
> [3] Tian et al., Multi-Agent Debate with Memory Masking, ICLR 2026

---

> > ### Author Rebuttal · Reviewer_yjbr · 2026-04-01
> >
> > Looking at the training curve results, was the experiment only conducted under one seed, without calculating the mean and standard deviation across different random seeds? This would be helpful for evaluating the robustness of the method.

---

> > > ### Author Response · Authors · 2026-04-01
> > >
> > > Dear reviewer, thanks for your follow-up question. To address this concern, we have added new figures (**[stage1_figure](https://anonymous.4open.science/r/hieramas-f350/stage1_multi_seeds.pdf) and [stage2_figure](https://anonymous.4open.science/r/hieramas-f350/stage2_multi_seeds.pdf)**) showing training curves over three independent runs with different random seeds. The figure now reports the mean and standard deviation across these three runs, providing a clearer assessment of robustness. Please feel free to let us know if you have any further questions or concerns.

---

### Official Review · Reviewer_sJA5 · 2026-03-11

**Soundness:** 2
**Presentation:** 3
**Significance:** 2
**Originality:** 2
**Overall Recommendation:** 4
**Confidence:** 4

**Summary:**

HieraMAS proposes a multi-agent framework where each agent role is composed of multiple heterogeneous LLMs working in a propose-then-synthesize structure, rather than a single model. The system jointly optimizes which LLMs to assign within each role, which roles to retain, and how roles communicate with one another. To handle the training challenges this creates, the authors introduce multi-level rewards for accurate per-agent feedback and reformulate topology selection as a graph classification task rather than edge-by-edge optimization. Experiments on coding, math, and general knowledge benchmarks show HieraMAS achieves state-of-the-art accuracy while remaining more cost-efficient than competing methods.

**Compliance With Llm Reviewing Policy:**

Affirmed.

**Key Questions For Authors:**

How does HieraMAS perform when restricted to only open-source models, given that reliance on GPT-5-Mini raises concerns about reproducibility and whether the gains are attributable to the framework or simply to model strength?

**Limitations:**

See Weakness section

**Strengths And Weaknesses:**

**Strengths**

- The two-stage training algorithm is well-motivated, with theoretical proofs supporting why per-edge optimization fails and why holistic graph classification is preferable
- HieraMAS makes a useful engineering and algorithmic contribution by jointly optimizing dimensions that prior MAS work treated separately, but it operates squarely within the existing MAS research tradition rather than reframing it.
- Experiments span three diverse benchmarks with multiple baselines, and cost comparisons are reported alongside accuracy

**Weaknesses**

- The generalization analysis (Table 3) reveals meaningful performance gaps on unseen domains, which the paper acknowledges but does not fully resolve
- The proposer-synthesizer structure within supernodes is adopted from prior work and not itself optimized, leaving a potentially important design choice unexplored
- Theoretical assumptions, particularly that LLM capability matters more than topology, are stated but not empirically verified in isolation
- Reproducibility may be limited given reliance on proprietary models like GPT-5-Mini that are not publicly accessible

---

> ### Author Rebuttal · Authors · 2026-03-31
>
> We sincerely thank Reviewer sJA5 for the detailed review and the recognition of our two-stage training design and cost-efficiency analysis. Our point-by-point responses are as follows.
>
> **W1: Generalization on unseen domains**
>
> We acknowledge that out-of-domain generalization poses a unique challenge for HieraMAS (and MASRouter), as both methods access training data to learn a **mixture of LLMs** per sample. Other baselines always use the strongest LLM, which inherently provides a generalization advantage without requiring any learned selection to transfer.
>
> The most direct comparison is with MASRouter. In Table 3 of our draft, HieraMAS outperforms MASRouter by **+12.00%** (GPT-5-Mini setting) and **+4.00%** (Qwen3-80B setting) on 5 unseen MMLU subjects, demonstrating the effectiveness of our two-stage design with multi-level rewards. Compared to AFlow and GDesigner, HieraMAS performs competitively (68.00% vs. 72.00% under GPT-5-Mini) despite using potentially weaker LLMs from the LLM pool.
>
> To further close this gap, we propose a lightweight **adaptation strategy**: finetuning only the Stage 2 GCN classifier with a few samples from unseen domains, while keeping the Stage 1 LLM selector frozen. Results (GPT-5-Mini setting):
>
> - **MATH**: Train on 5 problem types, test on 2 unseen types.
> - **MMLU-Redux**: Train on 25 subjects, test on 5 unseen subjects.
>
> Table: Performance comparison on unseen domains in MMLU-Redux
> | Method | MATH (Unseen) | MMLU-Redux (Unseen) |
> |--------|---------------|---------------------|
> | GDesigner | 92.50 | 72.00 |
> | MASRouter | 90.00 | 56.00 |
> | **Ours (Zero-shot)** | **97.50** | 68.00 |
> | **Ours (Few-shot Adaptation)** | **97.50** | **72.00** |
>
> On MATH, HieraMAS achieves **97.50%** zero-shot, outperforming GDesigner (+5.00%) and MASRouter (+7.50%). On MMLU-Redux, few-shot finetuning improves HieraMAS from 68.00% to **72.00%**, matching GDesigner. These results confirm that with further finetuning on communication patterns, HieraMAS is able to achieve better performance on unseen domains.
>
> **W2: Proposer-synthesizer structure not optimized**
>
> We agree with the reviewer this is a promising direction for improvement. In the current design, we fix the proposer-synthesizer structure to keep the optimization tractable, as jointly learning intra-supernode architecture alongside LLM selection and inter-node topology would significantly increase the search space complexity and training cost. We will explore learnable intra-supernode structures as future work.
>
> **W3: LLM capability vs. topology**
>
> To empirically verify this, we conduct an additional experiment restricting the LLM pool to **only weak models** (Qwen3-8B, Llama-3.1-8B-Instruct, Gemma-3-27B-IT) and running HieraMAS on MATH and MMLU-Redux:
>
> | Setting | MATH | MMLU-Redux |
> |---------|------|------------|
> | HieraMAS (weak-only pool) | 86.67 | 85.60 |
> | CoT (GPT-5-Mini) | 92.22 | 93.60 |
> | HieraMAS (Qwen3-80B setting) | 95.56 | 89.60 |
>
> With only weak LLMs, HieraMAS **underperforms** even single-agent CoT with GPT-5-Mini, confirming that multi-agent collaboration alone cannot compensate for insufficient LLM capability. However, once stronger models are added to the pool (Qwen3-80B setting), HieraMAS achieves **95.56%** and **89.60%**, surpassing CoT by +3.34% and closing the MMLU gap. This supports our claim that **LLM capability is a prerequisite**, while topology optimization provides further gains on top of capable models.
>
> **W4: Reproducibility with proprietary models**
>
> We clarify that HieraMAS interacts with all LLMs solely through **API calls** without modifying any model weights. As long as the OpenAI official API remains accessible, our experiments with GPT-5-Mini are fully reproducible. Furthermore, our Qwen3-80B setting uses an LLM pool consisting entirely of **open-source models**, providing a fully reproducible experimental setting that does not depend on any proprietary models. We will also open-source our code to further support reproducibility.
>
> ---
>
> We hope these responses address your concerns, and we would be happy to provide further clarification if needed.

---

### Official Review · Reviewer_uRC9 · 2026-03-12

**Soundness:** 3
**Presentation:** 2
**Significance:** 2
**Originality:** 2
**Overall Recommendation:** 4
**Confidence:** 3

**Summary:**

The paper proposes a hierarchical collaborative framework for MAS-LLM systems equipped with a communication system between nodes and the notion of supernode as a conglomerate of LLMs that solve tasks via a propose-synthesis approach. The paper further proposes a method to optimise a network’s behaviour by providing feedback to each node, as well as to each mixture/conglomerate of LLMs.
The optimisation mechanism aims at making the topology optimal, as well as the composition of each mixture of LLMs. The experiments are conducted on a series of standard benchmarks for math, coding, and language understanding.

**Compliance With Llm Reviewing Policy:**

Affirmed.

**Key Questions For Authors:**

Q1. My understanding is that each network is learnt based on a target task to solve. If that’s the case, isn’t it expensive to “train” the best topology for each specific subtask? Did the authors think of network adaptation to make this phase lighter (what I mean is that once a network is trained on a task, say math, one can adapt it to a similar benchmark without fully retraining it)?

Q2. Based on Q1, did the authors test how well a topology performs on a similar task without retraining the topology?

Q3. Exposing your network, at training time, to samples from the same distribution you use to then test them, may be an advantage the other models in your compassion do not have. Can the authors clarify if they tested the other methods before before or after fine-tuning them on the 80% of each target benchmark?

**Limitations:**

Figures (Fig.1 and 2, for example) are really confusionary and make it extremely hard to understand what your paper is about. I suggest to simplify them and keep only the essential to showcase what your paper does.

Your method exposes a network to the training set you successively test on (despite they do not overlap). My main concern is that a comparison with other methods that do not have access to this information is biased in favour of your method. I think a fair comparison would consists of fine-tuning the other models with the same train/test split you use in your training (see Appendix D1).

In this sense, I am not convinced by the results in the paper. Despite there is clear evidence that adding more agents improves the performance of a system (from classic results in ensembles to “More agents is all you need”), Table 1 clearly shows that your system is, in many cases (4/6) not the best performer, and not event the first or second best (2/6). These datasets are standard benchmarks for LLMs, and I suggest that since many LLMs memorise them, to find tasks where a MAS really helps (it can be a collaborative task, rather than a one-shot task that is turned into a collaborative one).

Further, I couldn’t find an analysis of the evaluation costs (tokens, for example), which I think would shed a different light on the experiments. If, for example, your system needs a x10 number of tokens, the fact that it is the average-best performer wouldn’t be surprising (especially considering that ensembles help with diverse tasks (I do not want to mention the no-free lunch, but that gives a perspective on why your system achieves the average best performance but rarely is the best performer)).

**Strengths And Weaknesses:**

The idea of optimising inter- and intra-node communication and collaboration is not novel in MAS and LLM-MAS systems (see “How to Train a Leader: Hierarchical Reasoning in Multi-Agent LLMs” from 2025, and “Modular Pluralism: Multi-LLM Collaboration for Diverse Outputs” from 2024, but also classic works from the 1980’ like the Contract Net Protocol). Furthermore, the idea of using reinforcement learning to train a network of heterogenous nodes is not novel as well (e.g., “Federated and Hierarchical Multi-Agent Reinforcement Learning”, from 2017).

With that being said, the paper proposes what is seems to me a solid framework to optimise a network topology and agents composition. I think the idea of optimising a network for a task, opposed to optimising each specific agent and then inserting them in a plausibly good network, is a pro of this paper and can open up to further research in this field.

I appreciate the fact that training your system is less intensive than other approaches like Full-graph and that this analysis is present in the paper. I also appreciate the authors report the top-learnt topologies. It would be interesting to study such topologies with static analyses from Complex Network Theory (and thus beyond the simple Jaccard similarity) to understand what brings them together (e.g., the in-out node degree) and which one is better at what task.

I also appreciate the ablation study in section 4.3, which shows that your learnt MAS network is indeed introducing favourable elements to achieve higher performance in a task.

---

> ### Author Rebuttal · Authors · 2026-03-31
>
> We thank Reviewer uRC9 for the constructive feedback and the appreciation of our framework design and ablation study. We address each point below.
>
> **Q1 & Q2: Training cost and cross-task transfer**
>
> We clarify that HieraMAS is trained on a collection of subtasks, not a single subtask. MATH covers 7 problem types, and MMLU-Redux covers 25 subject categories. The learned LLM selector and graph classifier generalize across these heterogeneous tasks within each domain.
>
> We evaluated **zero-shot transfer** to held-out unseen categories (Table 3 for MMLU-Redux) in the draft. We now supplement with MATH adaptation and transfer, with MMLU-Redux adaptation results. Notably, only **HieraMAS and MASRouter** perform learnable LLM selection, other methods always use the strongest LLM, giving them an implicit generalization advantage. Thus, zero-shot transfer test is more meaningful for HieraMAS and MASRouter.
>
> **Setup** : MATH, train on 5 subjects, test on remaining 2 unseen subjects; MMLU-Redux, train on 25 subjects, test on 5 unseen subjects.
>
> Table: Evaluation on zero-shot transfer and adapatation on unseen domains
> | Method | MATH (Unseen) | MMLU-Redux (Unseen) |
> |--------|---------------|---------------------|
> | GDesigner | 92.50 | **72.00** |
> | MASRouter | 90.00 | 56.00 |
> | **Ours (Zero-shot)** | **97.50** | 68.00 |
> | **Ours (Few-shot adaptation)** | **97.50** | **72.00** |
>
> On MATH, HieraMAS outperforms MASRouter by **+7.50%** and GDesigner by **+5.00%** under zero-shot setting. On MMLU-Redux, HieraMAS outperforms MASRouter by **+12.00%**. With finetuning of **only the Stage 2 GCN** in unseen domains, HieraMAS reaches **72.00%** on MMLU-Redux. This highlights the further advantage of adapatation on graph topology.
>
> **Q3: Fairness of comparison — training data exposure**
>
> We clarify that our experimental setup ensures a fair comparison. Although some baselines such as CoT, SC+CoT, Full-Graph, and LLM-Debate do not require training, all learning-based methods (AFlow, GDesigner, MASRouter, and HieraMAS) are trained on the **same training set** and evaluated on the **same held-out test set**. There is no data advantage for our method.
>
> **L1: Performance ranking and benchmark difficulty**
>
> Regarding per-setting rankings: HieraMAS selects LLMs from a heterogeneous pool including potentially weaker models, while most baselines exclusively use the **strongest LLM**. Despite this harder setup, HieraMAS achieves the best average performance across all settings.
>
> Regarding benchmark difficulty: we agree that exploring more challenging tasks is a valuable future direction. However, our benchmarks are already among the more rigorous variants in this line of work. HumanEval++ augments the original HumanEval (as used in many previous work) with **80x more test cases**, substantially reducing false positives. MMLU-Redux is a **decontaminated and disambiguated** subset that removes erroneous questions from MMLU. Furthermore, benchmark memorization would benefit baselines using the strongest LLM **more than** HieraMAS, which assigns weaker, less-memorized models to certain positions.
>
> **L2: Evaluation cost analysis**
>
> We have reported inference cost ($) in **Table 4** of the original paper. We use monetary cost rather than raw token counts because different LLMs have vastly different pricing, which is precisely a core motivation of HieraMAS: **adaptively selecting cheaper models** instead of always using the most expensive one. To address the reviewer's concern, we provide a more detailed breakdown on MMLU-Redux (GPT-5-Mini setting):
>
> Table: Token and cost analysis on MMLU-Redux
> | Method | Acc | Input Tokens | Output Tokens | Cost ($) |
> |--------|-----|-------------|---------------|----------|
> | Full-Graph | 94.40 | 26,922 | 12,896 | 0.033 |
> | GDesigner | 92.00 | 20,874 | 12,041 | 0.029 |
> | MASRouter | 88.33 | 3,041 | 2,297 | 0.001 |
> | Ours | 95.20 | 182,192 | 48,133 | 0.010 |
>
> While HieraMAS uses more tokens due to its multi-agent collaboration structure, its actual cost (**$0.010**) is **3.3x cheaper** than Full-Graph and **2.9x cheaper** than GDesigner. This is because HieraMAS learns to assign cheaper LLMs to appropriate positions, achieving the highest accuracy at a fraction of the cost. MASRouter achieves the lowest cost but at significantly lower accuracy (88.33% vs. 95.20%).
>
> ---
>
> We hope the above clarifications and additional results address the reviewer's concerns. We would be happy to provide further clarification if needed.

---

> > ### Author Rebuttal · Reviewer_uRC9 · 2026-04-01
> >
> > I thank the authors for the response and the additional information about the costs and the clarifications about the performance.
> > I think this paper has its merits, so I am happy to increase the score.

---

> > > ### Author Response · Authors · 2026-04-01
> > >
> > > We appreciate the reviewer for your positive feedback and recognition. We will incorporate these additional results and clarifications into our revised paper. Please feel free to let us know if you have any further questions.

---

### Decision · Program_Chairs · 2026-04-30

**Decision:**

Accept (regular)

**Comment:**

The paper proposes HieraMAS, a multi-agent framework that jointly optimizes the intra-node mixtures and inter-node topology. All reviewers acknowledge that such joint optimization of intra-node mixtures and inter-node topology for multi-agent systems is interesting and novel. The authors' rebuttal further addressed some of the concerns from the reviewers, leading to the final agreement that all three reviewers gave the score of 4. The paper, after a thorough revision incorporating the constructive suggestions from the reviewers, could be accepted by ICML if there is room.